# Why Do Multi-Agent LLM Systems Fail?

**Mert Cemri**[1*]   **Melissa Z. Pan**[1*]   **Shuyi Yang**[2*]   **Lakshya A Agrawal**[1]   **Bhavya Chopra**[1]
**Rishabh Tiwari**[1]   **Kurt Keutzer**[1]   **Aditya Parameswaran**[1]   **Dan Klein**[1]
**Kannan Ramchandran**[1]   **Matei Zaharia**[1]   **Joseph E. Gonzalez**[1]   **Ion Stoica**[1]
[1]UC Berkeley   [2]Intesa Sanpaolo   *Equal Contribution

## Abstract

Despite enthusiasm for Multi-Agent LLM Systems (MAS), their performance gains on popular benchmarks are often minimal. This gap highlights a critical need for a principled understanding of why MAS fail. Addressing this question requires systematic identification and analysis of failure patterns. We introduce `MAST-Data`, a comprehensive dataset of 1600+ annotated traces collected across 7 popular MAS frameworks. `MAST-Data` is the first multi-agent system dataset to outline the failure dynamics in MAS for guiding the development of better future systems. To enable systematic classification of failures for `MAST-Data`, we build the first **M**ulti-**A**gent **S**ystem Failure **T**axonomy (`MAST`). We develop `MAST` through rigorous analysis of 150 traces, guided closely by expert human annotators and validated by high inter-annotator agreement ($\kappa = 0.88$). This process identifies 14 unique modes, clustered into 3 categories: (i) system design issues, (ii) inter-agent misalignment, and (iii) task verification. To enable scalable annotation, we develop an LLM-as-a-Judge pipeline with high agreement with human annotations. We leverage `MAST` and `MAST-Data` to analyze failure patterns across models (GPT4, Claude 3, Qwen2.5, CodeLlama) and tasks (coding, math, general agent), demonstrating opportunities for improvement through better MAS design. Our analysis provides insights revealing that identified failures require more sophisticated solutions, highlighting a clear roadmap for future research. We publicly release our comprehensive dataset (`MAST-Data`), the `MAST`, and our LLM annotator to facilitate widespread research and development in MAS.[1] [2]

*"Happy families are all alike; each unhappy family is unhappy in its own way."* (Tolstoy [1])
*"Successful systems all work alike; each failing system has its own problems."* (Berkeley'25)

## 1 Introduction

Recently, Large Language Model (LLM) based agentic systems have gained significant attention in the AI community [2–4]. Building on this characteristic, multi-agent systems are increasingly explored in various domains, such as software engineering, drug discoveries, scientific simulations, and general-purpose agents [5–11]. In this study, we define an LLM-based **agent** as an artificial entity with prompt specifications (initial state), conversation trace (state), and ability to interact with the environments such as tool usage (action). A **multi-agent system** (**MAS**) is then defined as a collection of agents designed to interact through orchestration, enabling collective intelligence. MAS are structured to coordinate efforts, enabling task decomposition, performance parallelization, context isolation, specialized model ensembling, and diverse reasoning discussions [12–17].

Despite the increasing adoption of MAS, their performance gains often remain minimal compared to single-agent frameworks [18] or simple baselines like best-of-N sampling [19]. Our empirical

---

[1]https://github.com/multi-agent-systems-failure-taxonomy/MAST
[2]https://huggingface.co/datasets/mcemri/MAST-Data

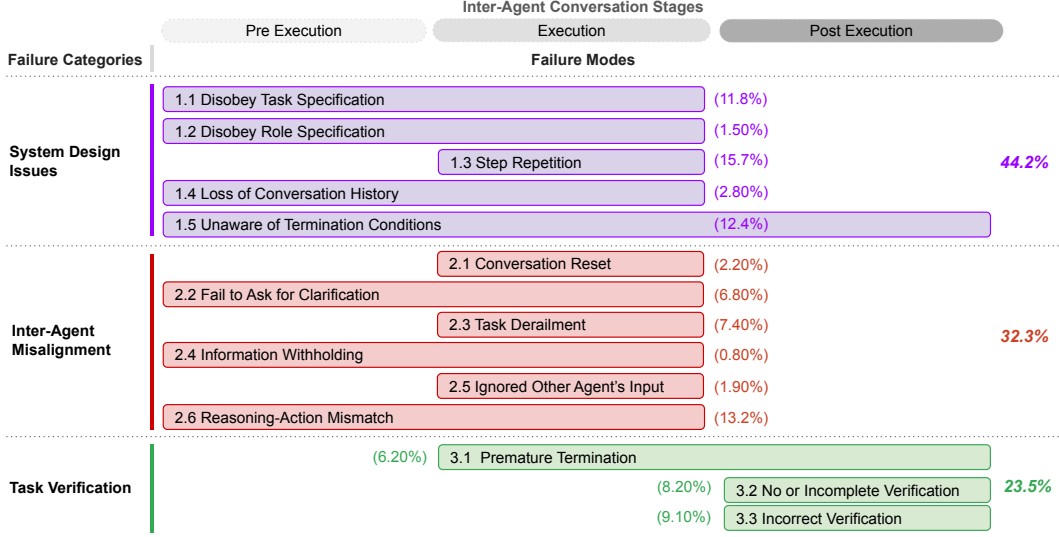

Figure 1: MAST: A **Taxonomy of MAS Failure Modes**. The inter-agent conversation stages indicate when a failure typically occurs within the end-to-end MAS execution pipeline. A failure mode spanning multiple stages signifies that the underlying issue can manifest or have implications across these different phases of operation. The percentages shown represent the prevalence of each failure mode and category as observed in our analysis of 1642 MAS execution traces. Detailed definitions for each failure mode and illustrative examples are available in Appendix A.

analysis reveals 41% to 86.7% failure rate on 7 state-of-the-art (SOTA) open-source MAS detailed in Figure 5 (Appendix B). Furthermore, there is no clear consensus on how to build robust and reliable MAS. This motivates the fundamental question we address: *Why do MAS fail?*

To address this question and systematically understand MAS failures, we introduce MAST-Data, a comprehensive, high-quality collection of 1642 annotated execution traces. We define failures as instances where the MAS does not achieve its intended task objectives. As shown in Table 1, we collect traces from 7 popular MAS frameworks run with two main model families (GPT-4 series and Claude series), covering tasks such as coding, math problem-solving, and general agent functionalities. Alongside MAST-Data, we also release MAST-Data-human, a smaller dataset featuring 21 traces annotated by three human experts each during our inter-annotator agreement studies. We create MAST-Data to outline failure dynamics in MAS and to guide the development of better future systems.

Systematically annotating failures in diverse MAS for a large-scale dataset like MAST-Data presents challenges unique to MAS: the difficulty in verifying ground truth for root cause detection and the absence of standardized failure definitions. To mitigate these challenges in creating MAST-Data, we first develop the **M**ulti-**A**gent **S**ystem Failure **T**axonomy (MAST), illustrated in Figure 1. We build MAST using Grounded Theory [20] from a close analysis of over 150 MAS execution traces (each averaging over 15,000 lines of text). This analysis spans a subset of five open-source MAS frameworks and involves six expert human annotators. To ensure generalizable definitions in MAST for labeling MAST-Data, three annotators independently and iteratively labeled a total of 15 traces until achieving high inter-annotator agreement ($\kappa = 0.88$). This comprehensive analysis results in 14 distinct failure modes, clustered into 3 categories. While MAST serves as a foundational first step towards unifying the understanding of MAS failures, we do not claim it covers every potential failure pattern. To enable scalable annotation for the full MAST-Data, we then develop an LLM-as-a-judge pipeline (which we term the LLM annotator) [21] using OpenAI's o1 model. We calibrate the LLM annotator to achieve high agreement with human expert annotations ($\kappa = 0.77$), and additionally validated applicability on two additional unseen MAS and benchmarks ($\kappa = 0.79$).

To demonstrate MAST's practical usage, our case studies (Appendix H) highlight its role in guiding MAS development, and in Section C we describe how to use the MAST easily as a python library using `pip install agentdash`. For example, the CPO agent in ChatDev can exhibit 'Failure Mode 1.2 - Disobey Role Specification' by terminating conversation without the CEO agent's consensus. We

demonstrate that a straightforward system workflow adjustment ensuring the CEO had the final say contributed to a +9.4% increase in overall task success rate. While such `MAST`-guided interventions demonstrate improvements, achieving robust MAS reliability often requires more than isolated fixes, pointing towards the need for more complex solutions and fundamental MAS redesigns.

Table 1: `MAST-Data` configuration details. HE: Human Evaluated (Task completions rates are checked by humans), HA: Human Annotated (Failure modes are annotated by humans), LA: LLM Annotated (Failure modes are annotated by LLM-as-a-Judge).

| MAS | Benchmark | LLM | Annotation | Trace # |
|---|---|---|---|---|
| ChatDev | ProgramDev | GPT-4o | HE, HA, LA | 30 |
| MetaGPT | ProgramDev | GPT-4o | HE, HA, LA | 30 |
| HyperAgent | SWE-Bench Lite | Claude-3.7-Sonnet | HE, HA, LA | 30 |
| AppWorld | Test-C | GPT-4o | HE, HA, LA | 30 |
| AG2 (MathChat) | GSM-Plus | GPT-4 | HE, HA, LA | 30 |
| Magentic-One | GAIA | GPT-4o | HE, HA, LA | 30 |
| OpenManus | ProgramDev | GPT-4o | HE, HA, LA | 30 |
| ChatDev | ProgramDev-v2 | GPT-4o | LA | 100 |
| MetaGPT | ProgramDev-v2 | GPT-4o | LA | 100 |
| MetaGPT | ProgramDev-v2 | Claude-3.7-Sonnet | LA | 100 |
| ChatDev | ProgramDev-v2 | Qwen2.5-Coder-32B-Instruct | LA | 100 |
| MetaGPT | ProgramDev-v2 | Qwen2.5-Coder-32B-Instruct | LA | 100 |
| ChatDev | ProgramDev-v2 | CodeLlama-7b-Instruct-hf | LA | 100 |
| MetaGPT | ProgramDev-v2 | CodeLlama-7b-Instruct-hf | LA | 100 |
| AG2 (MathChat) | OlympiadBench | GPT-4o | HE, LA | 206 |
| AG2 (MathChat) | GSMPlus | Claude-3.7-Sonnet | HE, LA | 193 |
| AG2 (MathChat) | MMLU | GPT-4o-mini | HE, LA | 168 |
| Magentic-One | GAIA | GPT-4o | HE, LA | 165 |

These findings suggest `MAST` reflects fundamental design challenges inherent in current MAS, not just artifacts of specific MAS implementation. By systematically defining failures, `MAST` serves as a framework to guide failure diagnosis and opens concrete research problems for the community. We have released our traces and annotations and open-sourced the LLM annotator pipeline to foster research in the design of more robust and reliable MAS.

The contributions of this paper are as follows:

- We introduce and **open-source** `MAST-Data`, the first large-scale MAS failure dataset with consistent annotations from 7 MAS and four model families. And `MAST-Data-human`, a detailed inter-annotator study results with human labels. Together serve to facilitate research into MAS failures.
- We introduce `MAST`, the first empirically grounded **taxonomy of MAS failures**, providing a structured framework for defining, understanding and annotating failures.
- We develop a scalable LLM-as-a-judge **annotation pipeline** integrated with `MAST` for efficiently annotating `MAST-Data` and enabling analysis of MAS performance, diagnosis of failure modes, and understanding of failure breakdowns.
- We demonstrate through **case studies** that failures identified by `MAST` often stem from system design issues, not just LLM limitations or simple prompt following, and require more than superficial fixes, thereby highlighting the need for structural MAS redesigns.

## 2 Related Work

### 2.1 Challenges in Agentic Systems

The promising capabilities of agentic systems have inspired research into solving specific challenges. For instance, Agent Workflow Memory [22] addresses long-horizon web navigation by introducing workflow memory. DSPy [23] tackles issues in programming agentic flows, while StateFlow [24] focuses on state control within agentic workflows to improve task-solving capabilities. Several surveys also highlight challenges and potential risks specifically within MAS [25, 26]. While these works meaningfully contribute towards understanding specific issues or providing high-level overviews, they do not offer a fine-grained, empirically grounded taxonomy of *why* MAS fail across diverse systems and tasks. Numerous benchmarks also exist to evaluate agentic systems [27–32]. These evaluations are crucial but primarily facilitate a top-down perspective, focusing on aggregate performance or high-level objectives like trustworthiness and security [33, 34]. Our work complements these efforts by providing a bottom-up analysis focused on identifying specific failure modes in MAS.

### 2.2 Design Principles for Agentic Systems

Several works highlight challenges in building robust agentic systems and suggest design principles, often focused on single-agent settings. For instance, Anthropic's blog post emphasizes modular components and avoiding overly complex frameworks [35]. Similarly, Kapoor et al. [19] demonstrates how complexity can hinder practical adoption. Our work extends these insights to the multi-agent context. By systematically collecting and analyzing a large corpus of MAS failure instances within `MAST-Data`, and by developing `MAST`, we provide not only a structured understanding of *why* MAS fail but also empirical data from `MAST-Data` to support the development and validation of more robust design principles for MAS. This aligns with the call for clearer specifications and design principles [36].

### 2.3 Related Datasets and Taxonomy

Despite the growing interest in LLM agents, dedicated research systematically characterizing their failure modes remains limited, particularly for MAS. While Bansal et al. [37] catalogs challenges in human-agent interaction, our contribution focuses specifically on failures within autonomous MAS execution. Other related work includes taxonomies for evaluating multi-turn LLM conversations [38] or specific capabilities like code generation [39]. These differ significantly from our goal of developing a generalizable failure taxonomy for multi-agent interactions and coordination.

Further related efforts aim to improve MAS through different approaches. AgentEval [40] proposes a framework using LLM agents to define and quantify multi-dimensional evaluation criteria reflecting task utility for end-users. AGDebugger [41] introduces an interactive tool enabling developers to debug and steer agent teams by inspecting and editing message histories. And currentwork by Zhang et al. [42] present the Who&When dataset and MAS debugger, which focuses on summarizing failures for specific task items by attributing them to particular agents and error steps.

Thus, `MAST-Data` and `MAST` represent, to our knowledge, the first empirically derived, comprehensive dataset and taxonomy focused specifically on MAS failures focus on failure patterns. Identifying these patterns highlights the need for continued research into robust evaluation metrics and mitigation strategies tailored for the unique challenges of MAS.

## 3 The Multi-Agent Systems Dataset

To facilitate a principled understanding of why MAS fail and to guide the development of more reliable future systems, we introduce `MAST-Data`, the Multi-Agent System Failure Dataset. `MAST-Data` is a comprehensive, empirically grounded dataset comprising 1642 annotated execution traces collected from 7 popular MAS frameworks, covering domains of coding, math, and generic tasks.

Constructing such a dataset, however, presents distinct challenges. First, unlike in traditional software where failures often have clearly identifiable root causes, failures in MAS are frequently complex. They involve convoluted agent interactions and the compounding effects of individual model behaviors and overall system design. Therefore, pinpointing the precise nature and origin of a failure in MAS

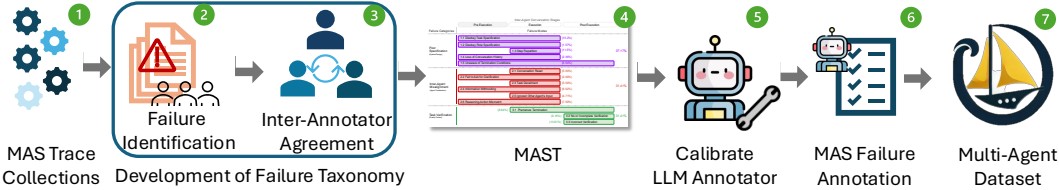

Figure 2: Methodological workflow for constructing the `MAST-Data` dataset, involving the empirical identification of failure modes, the development of `MAST`, iterative refinement through inter-annotator agreement studies ($\kappa = 0.88$), and the creation of a scalable LLM annotation pipeline. This figure highlights our systematic approach to creating a comprehensive dataset for studying MAS failures.

requires more than simple error detection; it necessitates understanding the system's dynamics. Second, the lack of a standardized failure framework with clear definitions makes identifying and classifying MAS failures across different systems inconsistent, which complicates annotation and cross-system analysis.

To address these challenges, we develop a rigorous, principled methodology to construct `MAST-Data`. In this section, we detail our approach, which centers on building the first empirical MAS failure taxonomy, `MAST`, and a scalable annotation pipeline for systematic and comprehensive data collection. Figure 2 summarizes our methodological workflow.

## 3.1 Data Collection with Grounded Theory Analysis

To uncover a comprehensive set of failure patterns that are both diverse and generalizable to standardize failure labels which we detail further in Section 3.2, we first collect 150 traces from five MAS frameworks, which are closely examined by six human experts. Our goal at this stage is to identify as many distinct failure modes as possible, ensuring these observed patterns are not merely artifacts of a single system but can likely apply more broadly. To achieve this without predefined hypotheses, we adopt the **Grounded Theory** (GT) approach [20]. This qualitative research method allows failure modes to emerge organically from empirical data.

For this initial data collection, we use *theoretical sampling* [43] to ensure robust coverage across different system objectives and interaction patterns. This method guides our selection of the five MAS frameworks (HyperAgent, AppWorld, AG2, ChatDev, and MetaGPT) and two task categories (programming and math problem-solving). We then iteratively analyze these traces using core GT techniques: *open coding* [44] to label trace data with observed failure behaviors; *constant comparative analysis* to refine our understanding of these failure behaviors and their recurrence across systems; *memoing* to document insights; and *theorizing* to structure these findings into an initial set of failure modes with their definitions. This iterative analysis continues until we reach *theoretical saturation*, where further data analysis does not yield new failure mode insights. This initial process requires significant human effort, over 20 hours of annotation per expert for these 150 traces.

## 3.2 Standardizing Failure Labels via Inter-Annotator Agreement

To make the failure observations from our GT analysis useful for creating consistent labels in `MAST-Data`, we recognize the critical need for standardized definitions that apply uniformly across different MAS. To address this, we develop the failure taxonomy - `MAST`. `MAST` serves as a foundational first step towards a common understanding of MAS failures by providing clear, empirically grounded failure observation labels. We provide a detailed description and analysis of `MAST` in Section 4.

To develop a taxonomy that is unambiguous and consistently applicable by different annotators, we rigorously validate and refine `MAST` definitions through Inter-Annotator Agreement (IAA) studies. This iterative process begins with a preliminary version of `MAST` derived from our GT findings. In each round of IAA, three expert annotators independently label a subset of five randomly selected traces from our initial 150+ trace collection using `MAST`. We then facilitate discussions to collectively resolve any disagreements. Based on these discussions, we iteratively refine `MAST` by adjusting failure mode definitions, adding new modes, or removing and merging existing ones until we achieve

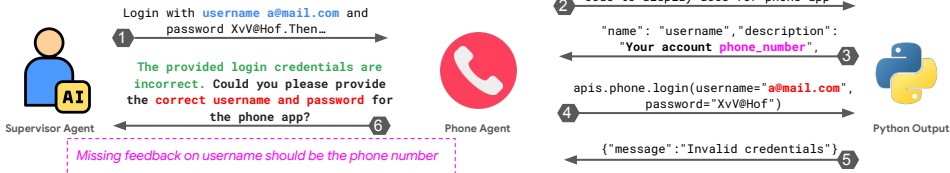

Figure 3: Visualization of a trace segment in `MAST-Data`. This illustrates an agent-to-agent conversation exhibiting Failure Mode 2.4: Information Withholding. The Phone Agent fails to communicate API requirements (username format) to the Supervisor Agent, who also fails to seek clarification, leading to repeated failed logins and task failure.

high consensus. We conduct three such rounds of IAA, requiring about 10 hours in total solely for resolving disagreements, not including the annotation time itself. We measure agreement using Cohen's Kappa score, achieving a strong average of $\kappa = 0.88$ in the final rounds. This high IAA score signifies that `MAST` provides a clear and shared understanding of failure modes, crucial for the consistent annotation of `MAST-Data`. Figure 3 illustrates an example of a trace snippet with a `MAST` label.

### 3.3 Enabling Scalable Annotation: The LLM-as-a-Judge Pipeline

Manually annotating over 1600 MAS traces with fine-grained failure modes is time-consuming and costly. To enable scalable and automated failure annotation for `MAST-Data`, we develop an LLM-as-a-Judge pipeline (LLM annotator), building upon our validated `MAST`. This pipeline prompts an LLM (OpenAI's o1 model) with an execution trace, the `MAST` definitions, and few-shot examples from our human-annotated data (details in Appendix N) to classify observed failure modes. We validate the LLM annotator's reliability against expert human annotations on a held-out set from our IAA studies. The LLM annotator achieves high agreement with human experts (accuracy 94%, Cohen's Kappa of 0.77; Table 2), confirming its suitability for scaling the annotation process while adhering to `MAST` definitions.

Table 2: Performance of LLM-as-a-judge pipeline

| Model | Accuracy | Recall | Precision | F1 | Cohen's $\kappa$ |
|---|---|---|---|---|---|
| o1 | 0.89 | 0.62 | 0.68 | 0.64 | 0.58 |
| o1 (few shot) | 0.94 | 0.77 | 0.833 | 0.80 | 0.77 |

### 3.4 Constructing the Multi-Agent Dataset

Before large-scale data collection for `MAST-Data`, we confirm the generalizability of our finalized `MAST` and the LLM annotator. We evaluate their performance on two new MAS (OpenManus and Magentic-One) with two new benchmarks (MMLU and GAIA, the latter representing a new general-agent task domain for validation) not part of the initial `MAST` development. An additional human IAA round on these out-of-domain traces using the finalized `MAST` yields a strong Cohen's Kappa score of 0.79. This demonstrates `MAST`'s effectiveness in capturing failures in diverse systems and tasks without further modification, supporting the robustness of our annotation approach for broader application. We further detail the uniqueness of `MAST` failure modes via a correlation study in Appendix E.

Leveraging our validated `MAST` and LLM annotator, we expand data collection to construct `MAST-Data`, comprising 1642 annotated traces from seven popular MAS frameworks (Table 1). These frameworks include the five from our initial studies, the two from the generalization validation, and Manus [45] as detailed in Appendix B. These traces cover diverse tasks like coding, math problem-solving, and general agent functionalities. For `MAST-Data`, our LLM annotator identifies `MAST` failure modes in each trace and provides a corresponding reason. We also release `MAST-Data-human`, consisting of all traces annotated by human experts during our IAA studies, where each annotation specifies `MAST` failure modes with textual justifications. We open-source

`MAST-Data` and `MAST-Data-human` as resources to analyze MAS failure dynamics and guide robust system design.

# 4 The Multi-Agent System Failure Taxonomy

This section details `MAST`, a key result of our study and a critical component that guides the creation and analysis of `MAST-Data`. `MAST` provides the first empirically grounded, structured framework for defining, understanding, and annotating common failures in MAS. Here, we present its structure, the failure categories it defines, and key insights derived from its development and application.

`MAST`, illustrated in Figure 1, identifies 14 fine-grained failure modes, which we map to MAS execution stages (Pre-Execution, Execution, and Post-Execution) where their root causes commonly emerge. These modes are organized into 3 overarching categories reflecting the fundamental nature of the observed failures. While we recognize that prior works have noted some individual failure types and we do not claim `MAST` is exhaustive, it offers precise definitions for a structured approach to understanding why MAS fail. Detailed definitions for each failure mode (FM) are available in Appendix A, with specific examples in Appendix N.

We acknowledge that some MAS failures can stem from fundamental limitations of current LLMs, such as hallucination or instruction following. However, in developing `MAST`, we focus on identifying failure patterns where improvements in system design, agent coordination, and verification can offer room to improve the reliability of MAS, often independently of or complementary to advancements in the base models themselves. We now discuss each failure category (FC) in `MAST` and its implications.

> **FC1. System Design Issues.** Failures originate from system design decisions, and poor or ambiguous prompt specifications.
> 💡 **Insight 1.** MAS failure is not merely a function of challenges in the underlying model; a well-designed MAS can result in performance gain when using the same underlying model.

Failures in FC1 occur during execution but often reflect flaws in pre-execution design choices regarding system architecture, prompt instructions, or state management. These include failing to follow task requirements (FM-1.1, 11.8%) or agent roles (FM-1.2, 1.5%), step repetitions (FM-1.3, 15.7%), context loss (FM-1.4, 2.80%), or not recognizing task completion (FM-1.5, 12.4%). While FM-1.1 and FM-1.2, disobey specifications, may seem like general instruction-following limitation, we identify deeper causes: (1) flaws in MAS design regarding agent roles and workflow, (2) poor user prompt specifications, or (3) limitations of the underlying LLM. We posit that a well-designed MAS should interpret high-level objectives with minimal but clear user input to mitigate the impact of points (2) and (3).

For instance, when ChatDev is tasked to create a Wordle game with the prompt `a standard wordle game by providing a daily 5-letter...`, the generated program uses a fixed word dictionary. Even with a more explicit prompt like `...without having a fixed word bank, and randomly select a new 5-letter word each day`, ChatDev still produces code with a fixed list and new errors. This suggests failures stem from the MAS's design for interpreting specifications. Our intervention studies (Appendix H) show that improving agent role specifications alone yields a +9.4% success rate increase for ChatDev with the same user prompt and LLM (GPT-4o).

> **FC2. Inter-Agent Misalignment.** Failures arise from a breakdown in critical information flow from inter-agent interaction and coordination during execution.
> 💡 **Insight 2.** Solutions focused on context or communication protocols are often insufficient for FC2 failures, which demand deeper 'social reasoning' abilities from agents.

FC2 covers failures in agent coordination. These include unexpected conversation resets (FM-2.1, 2.20%), proceeding with wrong assumptions instead of seeking clarification (FM-2.2, 6.80%), task derailment (FM-2.3, 7.40%), withholding crucial information (FM-2.4, 0.85%), ignoring other agents' input (FM-2.5, 1.90%), or mismatches between reasoning and action (FM-2.6, 13.2%). Figure 3 illustrates information withholding (FM-2.4). Diagnosing FC2 failures can be complex, as similar surface behaviors (e.g., missing information) can stem from different root causes like withholding (FM-2.4), ignoring input (FM-2.5), or context mismanagement (FM-1.4), underscoring the need for `MAST`'s fine-grained modes.

Recent system innovations, such as Model Context Protocol [46] and Agent to Agent [47], improve agent communication by standardizing message formats from different tool or agent providers. However, the errors we observe in FC2 occur even when agents within the same framework communicate using natural language. This signals a deeper agent interaction dynamic challenge: the collapse of 'theory of mind' [48], where agents fail to accurately model other agents' informational needs. Addressing this likely requires structural improvements to the content of agent messages or enhancing models' contextual reasoning and their capacity to infer other agents' informational needs, such as through targeted training, as base LLMs are generally not pre-trained for such nuanced inter-agent dynamics. Thus, robust solutions will likely involve a combination of improved MAS architecture and model-level advancements in communicative intelligence.

> **FC3. Task Verification.** Failures involve inadequate verification processes that fail to detect or correct errors, or premature termination of tasks.
> 💡 **Insight 3.** Multi-Level Verification is Needed. Current verifier implementations are often insufficient; sole reliance on final-stage, low-level checks is inadequate.

FC3 failures are related to the quality control of the final output, including premature termination (FM-3.1, 6.20%), no or incomplete verification (FM-3.2, 8.20%), or incorrect verification (FM-3.3, 9.10%). These highlight challenges in ensuring output correctness and reliability. Systems with explicit verifiers like MetaGPT and ChatDev generally show fewer total failures (Figure 4), indicating explicit checks help. However, the presence of a verifier is not a silver bullet, as overall MAS success rates can still be low. For example (FM-3.2), a ChatDev-generated chess program passes superficial checks (e.g., code compilation) but contains runtime bugs because it fails to validate against actual game rules, rendering the output unusable despite review phases.

During our GT analysis of MAS traces, we find that many existing verifiers perform only superficial checks, despite being prompted to perform thorough verification, such as checking if the code compiles or if there are leftover `TODO` comments. We posit that MAS development should take lessons from traditional software development where programmers test their code before committing. More rigorous verification is needed, such as using external knowledge, collecting testing output throughout generation, and multi-level checks for both low-level correctness and high-level objectives. We demonstrate this in an intervention study where adding a high-level task objective verification step to ChatDev yields a +15.6% improvement in task success on ProgramDev (details in Appendix H).

## 5 Towards better Multi-Agent LLM Systems

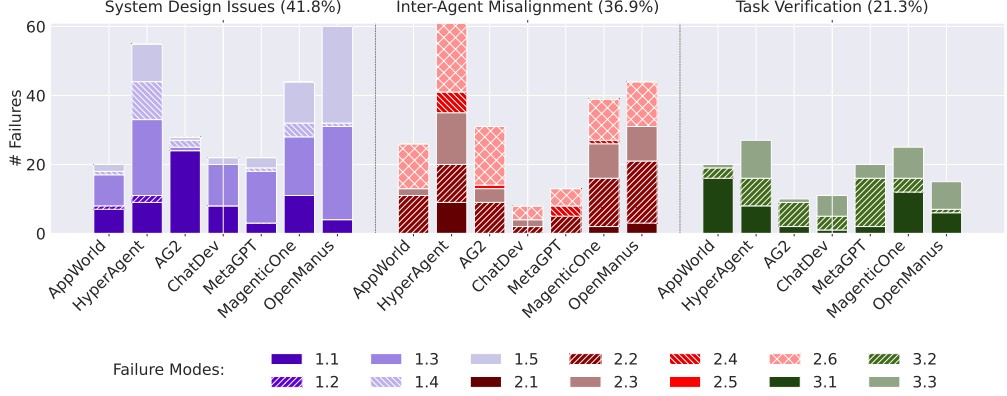

Figure 4: Distribution of failure in `MAST-Data` with `MAST` labels on total 210 traces. This plot visualizes the failure distributions of the first 30 traces for each system. As the specific tasks and benchmarks may differ across the MAS configurations shown, these results are intended to illustrate system-specific failure profiles rather than to serve as a performance comparison across MAS.

We now discuss the broader implications and usage of `MAST-Data` and `MAST`. MAST-Data, with its annotations grounded in `MAST`, provides crucial empirical evidence, while `MAST` offers a foundational framework and practical tool for understanding, debugging, and ultimately improving MAS. By

concretely defining failure modes and providing a large-scale dataset of their occurrences, our work outlines the challenges in building reliable MAS and opens targeted research problems for the community.

## 5.1    Failure Breakdown in `MAST-Data`

Our analysis of `MAST-Data` reveals that failure distributions differ markedly across various MAS, often reflecting their unique architectural characteristics and design philosophies. For example, as illustrated in Figure 4, we observe specific patterns: AppWorld frequently suffers from premature terminations (FM-3.1), potentially due to its star topology and lack of a predefined workflow making termination conditions less obvious; OpenManus exhibits a tendency towards step repetition (FM-1.3); and HyperAgent could benefit from addressing its dominant failure modes of step repetition (FM-1.3) and incorrect verification (FM-3.3). These system-specific profiles underscore that there is no one-size-fits-all solution to MAS failures.

We also use `MAST-Data` to study the impact of different underlying language models and MAS designs on failure patterns. For instance, when comparing GPT-4o and Claude 3.7 Sonnet within the MetaGPT framework on programming tasks, we find that while GPT-4o generally performs better than Claude, it shows significantly fewer FC1 (System Design Issues) failures by 39%. We also examine the impact of different MAS designs on the same benchmark, such as comparing MetaGPT and ChatDev on ProgramDev. Here, while MetaGPT generally outperforms ChatDev by having 60-68% less failure in FC1 and FC2, it has 1.56x more FC3 failure than ChatDev. These comparative analyses, detailed further in Appendix F, provide insights into how model choice and architectural patterns influence system performance and distribution of failures.

## 5.2    `MAST` as a Practical Development Tool

Developing robust MAS is challenging: aggregate success rates can obscure the specific impacts of optimizations. `MAST` addresses this by providing a structured vocabulary for systematic failure breakdown. Using our LLM annotator with `MAST`, developers can obtain quantitative analyses of failure profiles for specific systems. We demonstrate `MAST`'s practical usage in guiding MAS improvement in our case studies (Appendix H). The Failure Mode breakdown analysis (Appendix H.3) shows which failure modes were mitigated and reveals any resulting trade-offs. This granular view, moving beyond aggregate metrics, is crucial for understanding *why* an intervention works and for iterating effectively towards more robust systems.

## 5.3    Beyond Model Capabilities: The Primacy of System Design

While one could simply attribute failures in `MAST-Data` to limitations of present-day LLM (e.g., hallucinations, misalignment), we conjecture that improvements in the base model capabilities will be insufficient to address the full `MAST`. Instead, we argue that good MAS design requires organizational understanding – even organizations of sophisticated individuals can fail catastrophically [49] if the organization structure is flawed. Previous research in high-reliability organizations has shown that well-defined design principles can prevent such failures [50, 51].

Consistent with organization theories, our findings indicate that many MAS failures arise from the challenges in organizational design and agent coordination rather than the limitations of individual agents. In our intervention case studies (Appendix H), we apply MAS system workflow and prompt changes respectively (results in Table 5). With the same underlying model, we achieve max improvements of 15.6%. This highlights that MAS failures can be address with better system designs.

Although first step interventions lead to performance gains, not all failure modes are resolved, and task completion rates still remain low, indicating that more substantial improvements are needed. Achieving high reliability may requires combinatorial changes ranging from agent system organization to model level improvements (see Table 4). `MAST`, by providing a clear framework of failure points identified from `MAST-Data`, helps identify where these structural weaknesses lie and can guide the design and evaluation of more sophisticated MAS architectures.

# 6 Conclusion

In this study, we conduct the first systematic investigation into why MAS fail. This investigation results in the `MAST-Data`: a comprehensive public resource of over 1600 annotated execution traces from 7 popular MAS frameworks, which we create to outline MAS failure dynamics and guide future system development. To enable `MAST-Data`'s systematic annotation and analysis, we first develop the **M**ulti-**A**gent **S**ystem Failure **T**axonomy (`MAST`). We build `MAST` through a rigorous Grounded Theory-based analysis of an initial 150 traces, validating its definitions with strong inter-annotator agreement and identifying 14 distinct failure modes across 3 categories. For scalable annotation of `MAST-Data` using `MAST`, we then develop an LLM annotator, confirming its high agreement with human experts. Together, `MAST-Data` and `MAST` provide a foundational framework and empirical grounding for future MAS research.

We are excited about the potential of MAS, but their widespread adoption hinges on achieving greater reliability. Our work, through the public release of `MAST-Data`, `MAST`, and the LLM annotator, contributes towards this goal. `MAST-Data` offers a rich empirical basis for understanding current failure dynamics, while `MAST` provides a standardized language and framework to diagnose and mitigate these failures. By systematically identifying and categorizing challenges, we aim to open up concrete research directions and equip the community to develop more robust and effective multi-agent systems.

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

## Organization of Appendix

The appendix is organized as follows: in Section A further details about failure categories and failure modes are given, in Section B we provide some details about the multi-agent systems we have annotated and studied, in Section C we describe how to use the `MAST` easily as a python library using `pip install`, in Section D we describe the tasks in ProgramDev and ProgramDev-v2 Datsaet, in Section E we plot the correlations between MAS failure modes, in Section F we analyze the failure comparison between models and MAS, in Section G we discuss some tactical approaches and structural strategies to make MASs more robust to failures, in Section H we present two case studies where we show that tactical approaches can get only limited results, in Section I we present the failure mode distribution of the MAS frameworks powered by open-source language models, in Section J we present the correlations of failure mode distribution with some crucial statistics such as task completion rates and different benchmarks, in Section K we present the cost breakdown of LLM Annotator used in this paper for different MAS frameworks, in Sections L and M there are prompt interventions we tested on AG2 and ChatDev case studies, in Section N examples of every failure mode are reported and commented.

## A    `MAST` Failure Categories: Deep Dive

### A.1    FC1. System Design Issues

This category includes failures that arise from deficiencies in the design of the system architecture, poor conversation management, unclear task specifications or violation of constraints, and inadequate definition or adherence to the roles and responsibilities of the agents.

We identify five failure modes under this category:

- FM-1.1: **Disobey task specification** - Failure to adhere to the specified constraints or requirements of a given task, leading to suboptimal or incorrect outcomes.
- FM-1.2: **Disobey role specification** - Failure to adhere to the defined responsibilities and constraints of an assigned role, potentially leading to an agent behaving like another.
- FM-1.3: **Step repetition** - Unnecessary reiteration of previously completed steps in a process, potentially causing delays or errors in task completion.
- FM-1.4: **Loss of conversation history** - Unexpected context truncation, disregarding recent interaction history and reverting to an antecedent conversational state.
- FM-1.5: **Unaware of termination conditions** - Lack of recognition or understanding of the criteria that should trigger the termination of the agents' interaction, potentially leading to unnecessary continuation.

### A.2    FC2. Inter-Agent Misalignment

This category includes failures arising from ineffective communication, poor collaboration, conflicting behaviors among agents, and gradual derailment from the initial task.

We identify six failure modes under this category:

- FM-2.1: **Conversation reset** - Unexpected or unwarranted restarting of a dialogue, potentially losing context and progress made in the interaction.
- FM-2.2: **Fail to ask for clarification** - Inability to request additional information when faced with unclear or incomplete data, potentially resulting in incorrect actions.
- FM-2.3: **Task derailment** - Deviation from the intended objective or focus of a given task, potentially resulting in irrelevant or unproductive actions.
- FM-2.4: **Information withholding** - Failure to share or communicate important data or insights that an agent possess and could impact decision-making of other agents if shared.
- FM-2.5: **Ignored other agent's input** - Disregarding or failing to adequately consider input or recommendations provided by other agents in the system, potentially leading to suboptimal decisions or missed opportunities for collaboration.

- FM-2.6: **Reasoning-action mismatch** - Discrepancy between the logical reasoning process and the actual actions taken by the agent, potentially resulting in unexpected or undesired behaviors.

### A.3   FC3. Task Verification

This category includes failures resulting from premature execution termination, as well as insufficient mechanisms to guarantee the accuracy, completeness, and reliability of interactions, decisions, and outcomes.

We identify three failure modes under this category:

- FM-3.1: **Premature termination** - Ending a dialogue, interaction or task before all necessary information has been exchanged or objectives have been met, potentially resulting in incomplete or incorrect outcomes.

- FM-3.2: **No or incomplete verification** - (partial) omission of proper checking or confirmation of task outcomes or system outputs, potentially allowing errors or inconsistencies to propagate undetected.

- FM-3.3: **Incorrect verification** - Failure to adequately validate or cross-check crucial information or decisions during the iterations, potentially leading to errors or vulnerabilities in the system.

# B  Details of Multi-Agent Systems Evaluated

In this section, we provide details on MAS we evaluated during this study and their performance benchmark evaluation.

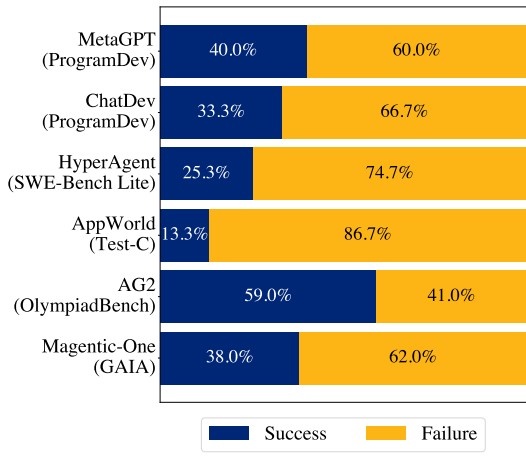

Figure 5: Failure rates of six popular Multi-Agent LLM Systems with GPT-4o and Claude-3.7-Sonnet. Performances are measured on different benchmarks, therefore they are not directly comparable.

## B.1  Overview of MAS

In this study, we evaluated 7 open-source frameworks. The architecture and the purpose of the systems is detailed in the table below.

Table 3: Overview of MAS covered in `MAST-Data`

| MAS | Agentic Architecture | Purpose of the System |
|---|---|---|
| MetaGPT [52] | Assembly Line | Simulating the SOPs of different roles in Software Companies to create open-ended software applications |
| ChatDev [5] | Hierarchical Workflow | Simulating different Software Engineering phases like (design, code, QA) through simulated roles in a software engineering company |
| HyperAgent [53] | Hierarchical Workflow | Simulating a software engineering team with a central Planner agent coordinating with specialized child agents (Navigator, Editor, and Executor) |
| AppWorld [54] | Star Topology | Tool-calling agents specialized to utility services (ex: Gmail, Spotify, etc.) being orchestrated by a supervisor to achieve cross-service tasks |
| AG2 [55] | N/A - Agentic Framework | An open-source programming framework for building agents and managing their interactions. |
| Magentic-One [11] | Star Topology | A generalist multi-agent system designed to autonomously solve complex, open-ended tasks involving web and file-based environments across various domains. |
| OpenManus [10] | Hierarchical | An open-source multi-agent framework designed to facilitate the development of collaborative AI agents that solve real-world tasks. It was inspired by the Manus AI agent. |

## B.2  Multi-Agent Systems in the Initial Annotation Phase

***MetaGPT.*** MetaGPT [52] is a multi-agent system that simulates a software engineering company and involves agents such as a Coder and a Verifier. The goal is to have agents with domain-expertise (achieved by encoding Standard Operating Procedures of different roles into agents prompts) collaboratively solve a programming task, specified in natural language.

***ChatDev.*** ChatDev is a generalist multi-agent framework that initializes different agents, each assuming common roles in a software-development company [56]. The framework breaks down the process of software development into 3 phases: design, coding and testing. Each phase is divided into sub-tasks, for example, testing is divided into code review (static) and system testing (dynamic). In every sub-task, two agents collaborate where one of the agents acts as the orchestrator and initiates the interaction and the other acts as an assistant to help the orchestrator achieve the task. The 2 agents then hold a multi-turn conversation to achieve the goal stated by the orchestrator ultimately leading to the completion of the task, marked by a specific sentinel by either agents. ChatDev has the following agent roles: CEO, CTO, Programmer, Reviewer and Tester. ChatDev introduces "Communicative Dehallucination", which encourages the assistant to seek further details about the task over multiple-turns, instead of responding immediately.

***HyperAgent.*** HyperAgent [53] is a framework for software engineering tasks organized around four primary agents: Planner, Navigator, Code Editor, and Executor. These agents are enhanced by specialized tools, designed to provide LLM-interpretable output. The Planner communicates with child agents via a standardized message format with two fields: Context (background and rationale) and Request (actionable instructions). Tasks are broken down into subtasks and published to specific queues. Child agents, such as Navigator, Editor, and Executor instances, monitor these queues and process tasks asynchronously, enabling parallel execution and significantly improving scalability and efficiency. For example, multiple Navigator instances can explore different parts of a large codebase in parallel, the Editor can apply changes across multiple files simultaneously, and the Executor can run tests concurrently, accelerating validation.

***AppWorld.*** AppWorld is a benchmark, that provides an environment with elaborate mocks of various everyday services like eShopping Website, Music Player, Contacts, Cost-sharing app, e-mail, etc [54]. The benchmark consists of tasks that require executing APIs from multiple services to achieve the end-users tasks. The AppWorld benchmark provides a ReAct based agent over GPT-4o as a strong baseline. We create a multi-agent system over AppWorld derived from the baseline ReAct agent, where each agent specializes in using one of the services mocked in AppWorld, with detailed instructions about the APIs available in that service, and access to the documentation for that specific service. A supervisor agent receives the task instruction to be completed, and can hold one-on-one multi-turn conversations with each of the service-specific agents. The service-agents are instructed to seek clarification with the supervisor, whenever required. The supervisor agent holds access to various information about the human-user, for example, credentials to access various services, name, email-id and contact of the user, etc, which the service-agents need to access the services, and must clarify with the supervisor agent.

***AG2.*** AG2 (formerly AutoGen) [57] is an open-source programming framework for building agents and managing their interactions. With this framework, it is possible to build various flexible conversation patterns, integrating tools usage and customizing the termination strategy.

## B.3 Closed-Source MAS

In our efforts to build a comprehensive dataset, we also explore popular closed-source platforms that are speculated to function as MAS. A notable example is Manus [45], a general AI agent platform. However, evaluating and incorporating such systems into `MAST-Data` for fine-grained failure analysis presents significant challenges. Specifically, with systems like Manus, the underlying language model is often not disclosed, and more critically, the platforms may not provide access to full agent execution traces. This lack of transparency into the internal conversational and operational steps makes reliable, detailed failure annotation using `MAST` infeasible. While we conduct human evaluation of task correctness for some closed-source systems (for instance, Manus achieves a 60% success rate on our ProgramDev benchmark), the absence of comprehensive trace data prevents their inclusion in the primary `MAST-Data` which focuses on deeply annotated failure dynamics. Our focus for `MAST-Data` thus remains on systems where such trace analysis can yield robust insights.

# C  `MAST` **Python Library**

In order to ease the usage of `MAST`, we also package our code as a pip installable python library, under the name `agentdash`. The example usage is shown below.

```
# Install the package (if you're in a notebook)
!pip install agentdash

from agentdash import annotator

# Initialize the annotator with your OpenAI API key
openai_api_key = "your-api-key"
MASTAnnotator = annotator(openai_api_key)

# Annotate a multi-agent system trace
trace = """
Agent1: I need to calculate the sum of 1 + 1.
Agent2: I'll help you with that. The answer is 3.
Agent1: Thank you! Task completed.
"""

mast_annotation = MASTAnnotator.produce_taxonomy(trace)

# View results
print("Failure Modes Detected:")
for failure_mode_id, detected in mast_annotation["failure_modes"].items():
    if detected:
        info = MASTAnnotator.get_failure_mode_info(failure_mode_id)
        print(f" {failure_mode_id}: {info['name']}")

print(f"\nSummary: {mast_annotation['summary']}")
print(f"Task Completed: {mast_annotation['task_completion']}")
print(f"Total Failures: {mast_annotation['total_failures']}")
```

# D  **ProgramDev and ProgramDev-v2 Datasets**

The ProgramDev dataset contains 30 coding problems [3]. These tasks are programming challenges, such as implementing Tic-Tac-Toe, Chess, or Sudoku, for which abundant solutions and descriptions are readily available online. We design ProgramDev with tasks intended to be relatively straightforward for MAS, rather than exceptionally difficult, to better isolate specific failure dynamics. We later extend this to ProgramDev-v2, a 100-problem dataset developed primarily for the comparative analyses of MAS architectures and underlying LLMs presented in Figure 8.

---

[3]`https://github.com/multi-agent-systems-failure-taxonomy/MAST/blob/main/traces/`
`programdev/programdev_dataset.json`

# E MAS Failure Modes Correlation

We evaluate MAST's effectiveness based on three key aspects: its generalization to unseen systems and datasets, the balanced distribution of identified failures, and the distinctiveness of its failure categories. This section details the correlation analysis.

Figure 6 shows low correlations (0.17-0.32). This suggests that the categories capture distinct aspects of MAS failures with limited overlap, supporting the taxonomy's structure. This distinctiveness is crucial because, as noted in Insight 2, failures with similar surface behaviors can stem from different root causes (e.g., memory management vs. agent coordination).

Although MAST's fine-grained nature helps differentiate root cause, it also poses a challenge for our LLM annotator. Analyzing correlations between specific failure modes (see Appendix E for Figure 7) shows moderate correlations (max of 0.63) between modes with similar symptoms might lead automated evaluators to conflate distinct root causes.

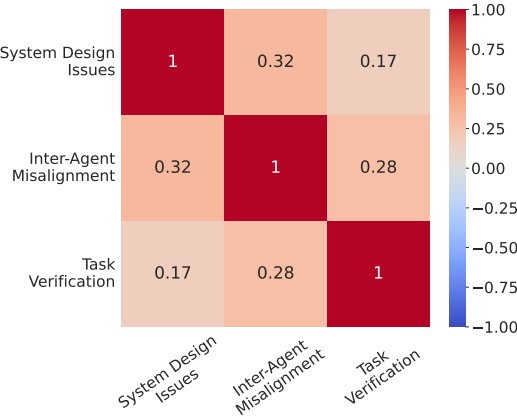

Figure 6: MAS failure modes correlation matrix

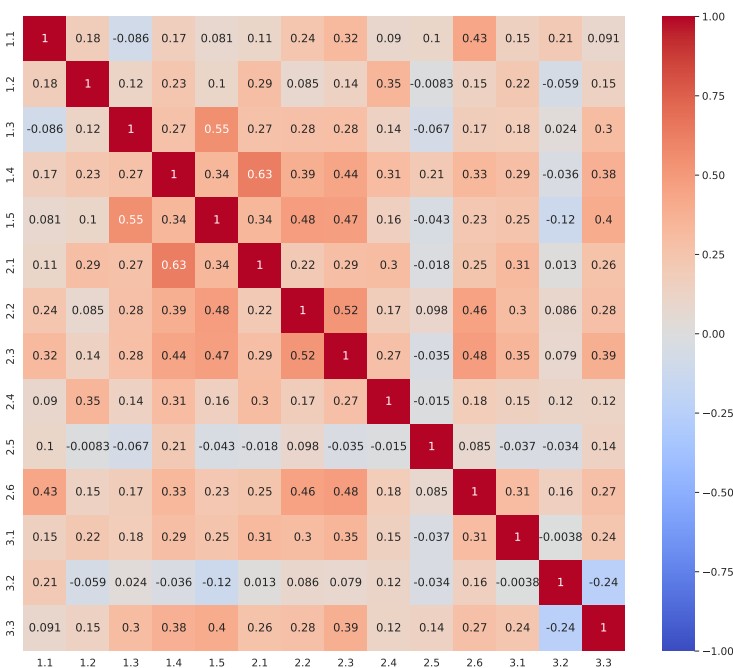

Figure 7: MAS failure modes correlation matrix

**Failure Mode Distribution Comparison: Effect of Underlying LLM**

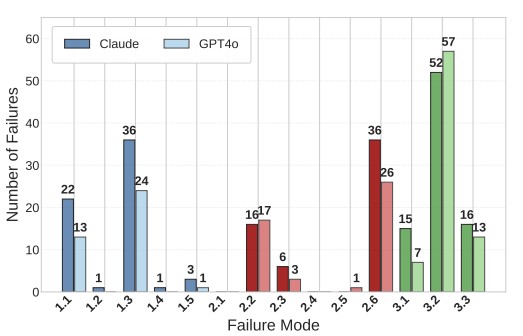 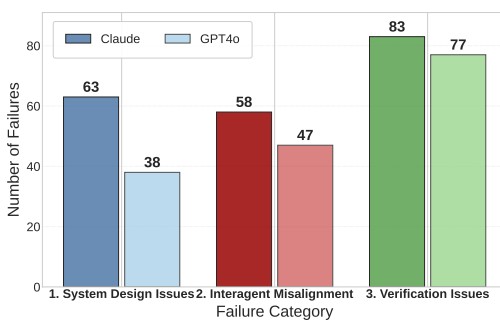

Figure 8: Comparison on `MAST` failure modes and categories on ProgramDev-v2 dataset explained in Section D to analyze LLM choice effect. MetaGPT is used for both cases with GPT-4o and Claude-3.7-Sonnet on two comparative cases.

**Failure Mode Distribution Comparison: Effect of MAS Framework**

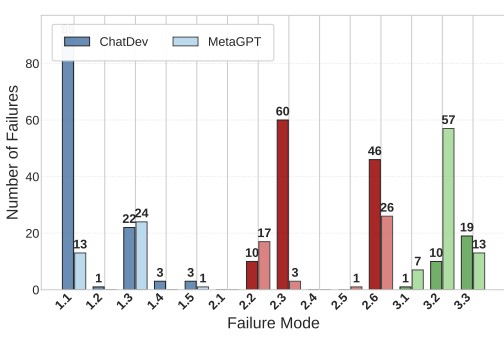 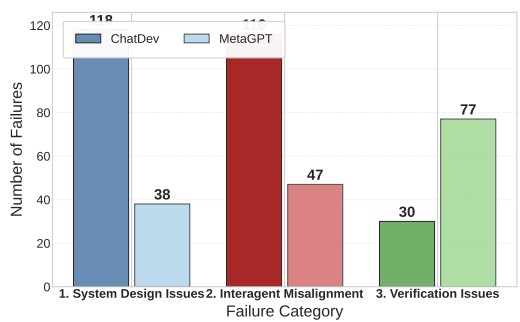

Figure 9: Comparison on `MAST` failure modes and categories on ProgramDev-v2 dataset explained in Section D to analyze MAS architecture effect. GPT-4o is used on two comparative cases, one using ChatDev and the other on MetaGPT.

# F   Understanding Failures: The Impact of Different LLMs and Agent Architectures

To understand how choices of underlying LLMs and MAS architectures influence failure patterns, we analyze results from our `MAST-Data`, categorized by `MAST` in the Figures 8 and 9.

First, we examine the impact of different LLMs by comparing GPT-4o and Claude 3.7 Sonnet within the MetaGPT framework on programming tasks Figure 8. Our findings indicate that GPT-4o exhibits substantially fewer failures in FC1 (System Design Issues, e.g., disobeying task or role specifications) and FC2 (Inter-Agent Misalignment, e.g., issues in coordination or communication) compared to Claude 3.7 Sonnet. This suggests GPT-4o may possess stronger capabilities in instruction following or aspects of 'social reasoning' for agentic collaboration within this setup. However, both models show a high number of failures in FC3 (Task Verification), indicating that robust verification remains a significant challenge regardless of the LLM used, though GPT-4o has a marginally lower count here.

Next, we investigate the effect of MAS architecture by comparing MetaGPT and ChatDev, both using GPT-4o as the underlying LLM, on the ProgramDev-v2 benchmark in Figure 9. We observe distinct failure profiles: MetaGPT demonstrates significantly fewer failures in FC1 (System Design Issues) and FC2 (Inter-Agent Misalignment) compared to ChatDev. This could imply that MetaGPT's architecture or operational flow is more effective at maintaining adherence to specifications and ensuring smoother agent coordination with GPT-4o. Interestingly, despite its stronger performance in FC1 and FC2, MetaGPT exhibits a considerably higher number of FC3 (Task Verification) failures than ChatDev. This may stem from the fact that in MetaGPT, the adherence to task specifications and

role specifications ar done mostly through SoPs, demonstrating strong performance in FC1 especially. However ChatDev places a higher importance in verification as it is reflected by the specific testing and reviewing phases in ChatDev's archiectural design, causing fewer verification issues. These results show that both the choice of LLM and the specific design of the MAS architecture critically shape the landscape of potential failures, and improvements likely require a holistic approach considering both aspects.

# G  Approaches and strategies to improve MASs

In this section, we discuss some approaches to make MASs more robust to failures. We categorize these strategies into two main groups: (i) **tactical approaches**, (ii) **structural strategies**. Tactical approaches involve straightforward modifications tailored for specific failure modes, such as improving the prompts, topology of the network of agents, and conversation management. In Section H, we experiment with such approaches in two case studies, and demonstrate that the effectiveness of these methods is not consistent. This leads us to consider a second category of strategies that are more comprehensive methods with system-wide impacts: strong verification, enhanced communication protocols, uncertainty quantification, and memory and state management. These strategies require more in-depth study and meticulous implementation, and remain open research topics for future exploration. See Table 4 for our proposed mapping between different solution strategies and the failure categories.

## G.1  Tactical Approaches

This category includes strategies related to improving prompts and optimizing agent organization and interactions. The prompts of MAS agents should provide clear description of instructions, and the role of each agent should be clearly specified (see L.2 as an example) [58, 59]. Prompts can also clarify roles and tasks while encouraging proactive dialogue. Agents can re-engage or retry if inconsistencies arise, as shown in Appendix L.5 [60]. After completing a complex multi-step task, add a self-verification step to the prompt to retrace the reasoning by restating solutions, checking conditions, and testing for errors [61]. However, it may miss flaws, rely on vague conditions, or be impractical [36]. Moreover, clear role specifications can be reinforced by defining conversation patterns and setting termination conditions [55, 62]. A modular approach with simple, well-defined agents, rather than complex, multitasked ones, enhances performance and simplifies debugging [63]. The group dynamics also enable other interesting possibilities of multi-agent systems: different agents can propose various solutions [64], discuss their assumptions, and findings (cross-verifications) [65]. For instance, in [66], a multi-agent strategy simulates the academic peer review process to catch deeper inconsistencies. Another set of tactical approaches for cross verifications consist in multiple LLM calls with majority voting or resampling until verification [67, 68]. However, these seemingly straightforward solutions often prove inconsistent, echoing our case studies' findings. This underscores the need for more robust, structural strategies, as discussed in the following sections.

## G.2  Structural Strategies

Apart from the tactical approaches we discussed above, there exist a need for more involved solutions that will shape the structure of the MAS at hand. We first observe the critical role of verification processes and verifier agents in multi-agent systems. Our annotations reveal that weak or inadequate verification mechanisms were a significant contributor to system failures. While unit test generation aids verification in software engineering [69], creating a universal verification mechanism remains challenging. Even in coding, covering all edge cases is complex, even for experts. Verification varies by domain: coding requires thorough test coverage, QA demands certified data checks [70], and reasoning benefits from symbolic validation [71]. Adapting verification across domains remains an ongoing research challenge.

A complementary strategy to verification is establishing a standardized communication protocol [72]. LLM-based agents mainly communicate via unstructured text, leading to ambiguities. Clearly defining intentions and parameters enhances alignment and enables formal coherence checks during and after interactions. [73] introduce Multi-Agent Graph Attention, leveraging a graph attention mechanism to model agent interactions and enhance coordination. Similarly, [74] propose Attentional Communication, enabling agents to selectively focus on relevant information. Likewise, [75] develop a learned selective communication protocol to improve cooperation efficiency.

Another important research direction is fine-tuning MAS agents with reinforcement learning. Agents can be trained with role-specific algorithms, rewarding task-aligned actions and penalizing inefficiencies. MAPPO [76] optimizes agents' adherence to defined roles. Similarly, SHPPO [77] uses a latent network to learn strategies before applying a heterogeneous decision layer. Optima [78] further enhances communication efficiency and task effectiveness through iterative reinforcement learning.

On a different note, incorporating probabilistic confidence measures into agent interactions can significantly enhance decision-making and communication reliability. Drawing inspiration from the framework proposed by Horvitz et al. [79], agents can be designed to take action only when their confidence exceeds a predefined threshold. Conversely, when confidence is low, agents can pause to gather additional information. Furthermore, the system could benefit from adaptive thresholding, where confidence thresholds are dynamically adjusted.

Although often seen as a single-agent property, memory and state management are crucial for multi-agent interactions, which can enhance context understanding and reduces ambiguity in communication. However, most research focuses on single-agent systems. MemGPT [3] introduces OS-inspired context management for an extended context window, while TapeAgents [80] use a structured, replayable log ("tape") to iteratively document and refine agent actions, facilitating dynamic task decomposition and continuous improvement.

Table 4: Solution Strategies vs. Failure Category in Multi-Agent Systems

| Failure Category | Tactical Approaches | Structural Strategies |
|---|---|---|
| System Design Issues | Clear role/task definitions, Engage in further discussions, Self-verification, Conversation pattern design | Comprehensive verification, Confidence quantification |
| Inter-Agent Misalignment | Cross-verification, Conversation pattern design, Mutual disambiguation, Modular agents design | Standardized communication protocols, Probabilistic confidence measures |
| Task Verification | Self-verification, Cross-verification, Topology redesign for verification | Comprehensive verification & unit test generation |

## H  Intervention Case Studies

In this section, we present the two case studies where we apply some of the tactical approaches. We also present the usage of MAST as a debugging tool, where we measure the failure modes in the system before applying any of the interventions, and then after applying the interventions we discuss below, and show that MAST can guide the intervention process as well as capture the improvements of augmentations.

### H.1  Case Study 1: AG2 - MathChat

In this case study, we use the MathChat scenario implementation in AG2 [57] as our baseline, where a Student agent collaborates with an Assistant agent capable of Python code execution to solve problems. For benchmarking, we randomly select 200 exercises from the GSM-Plus dataset [81], an augmented version of GSM8K [82] with various adversarial perturbations. The first strategy is to improve the original prompt with a clear structure and a new section dedicated to the verification. The detailed prompts are provided in Appendices L.1 and L.2. The second strategy refines the agent configuration into a more specialized system with three distinct roles: a Problem Solver who solves the problem using a chain-of-thought approach without tools (see Appendix L.3); a Coder who writes and executes Python code to derive the final answer (see Appendix L.4); a Verifier who reviews the discussion and critically evaluate the solutions, either confirming the answer or prompting further debate (see Appendix L.5). In this setting, only the Verifier can terminate the conversation once a solution is found. See Appendix L.6 for an example of conversation in this setting. To assess the effectiveness of these strategies, we conduct benchmarking experiments across three configurations (baseline, improved prompt, and new topology) using two different LLMs (GPT-4 and GPT-4o). We also perform six repetitions to evaluate the consistency of the results. Table 5 summarizes the results. The second column of Table 5 show that with GPT-4, the improved prompt with verification significantly outperforms the baseline. However, the new topology does not yield the same improvement. A Wilcoxon test returned a p-value of 0.4, indicating the small gain is not statistically significant. With GPT-4o (the third column of Table 5), the Wilcoxon test yields a p-value of 0.03 when comparing the baseline to both the improved prompt and the new topology, indicating statistically significant improvements. These results suggest that refining prompts and defining clear

agent roles can reduce failures. However, these strategies are not universal, and their effectiveness varies based on factors such as the underlying LLM.

## H.2 Case Study 2: ChatDev

ChatDev [5] simulates a multiagent software company where different agents have different role specifications, such as a CEO, a CTO, a software engineer and a reviewer, who try to collaboratively solve a software generation task. In an attempt to address the challenges we observed frequently in the traces, we implement two different interventions. Our first solution is refining role-specific prompts to enforce hierarchy and role adherence. For instance, we observed cases where the CPO prematurely ended discussions with the CEO without fully addressing constraints. To prevent this, we ensured that only superior agents can finalize conversations. Additionally, we enhanced verifier role specifications to focus on task-specific edge cases. Details of these interventions are in Section M. The second solution attempt involved a fundamental change to the framework's topology. We modified the framework's topology from a directed acyclic graph (DAG) to a cyclic graph. The process now terminates only when the CTO agent confirms that all reviews are properly satisfied, with a maximum iteration cutoff to prevent infinite loops. This approach enables iterative refinement and more comprehensive quality assurance. We test our interventions in two different benchmarks. The first one of them is a custom generated set of 32 different tasks (which we call as ProgramDev-v0, which consists of slightly different questions than the ProgamDev dataset we discussed in Section 4) where we ask the framework to generate programs ranging from "Write me a two-player chess game playable in the terminal" to "Write me a BMI calculator". The other benchmark is the HumanEval task of OpenAI. We report our results in Table 5. Notice that even though our interventions are successful in improving the performance of the framework in different tasks, they do not constitute substantial improvements, and more comprehensive solutions as we lay out in Section G.2 are required.

Table 5: Case Studies Accuracy Comparison. This table presents the performance accuracies (in percentages) for various scenarios in our case studies. The header rows group results by strategy: AG2 and ChatDev. Under AG2, GSM-Plus results are reported using GPT-4 and GPT-4o; under ChatDev, results for ProgramDev and HumanEval are reported. Each row represents a particular configuration: baseline implementation, improved prompts, and a redesigned agent topology.

| Configuration | AG2 | | ChatDev | |
|---|---|---|---|---|
| | GSM-Plus (w/ GPT-4) | GSM-Plus (w/ GPT-4o) | ProgramDev-v0 | HumanEval |
| Baseline | $84.75 \pm 1.94$ | $84.25 \pm 1.86$ | 25.0 | 89.6 |
| Improved prompt | $89.75 \pm 1.44$ | $89.00 \pm 1.38$ | 34.4 | 90.3 |
| New topology | $85.50 \pm 1.18$ | $88.83 \pm 1.51$ | 40.6 | 91.5 |

## H.3 Effect of the interventions on `MAST`

After carrying out the aforementioned interventions, we initially inspect the task completion rates as in Table 5. However, `MAST` offers us the opportunity to look beyond the task completion rates, and we can investigate the effects of these interventions on the failure mode distribution on these MASs (AG2 and ChatDev). As illustrated in Figures 10 and 11, we observe that both of these interventions cause a decrease across the different failure modes observed, and it is possible to conclude that topology-based changes are more effective than prompt-based changes for both systems. Moreover, this displays another usage of `MAST`, which is as well as an analysis tool after execution, it can serve as a debugging tool for future improvements as it shows which failure modes particular augmentations to the system can solve or miss, guiding future intervention decisions.

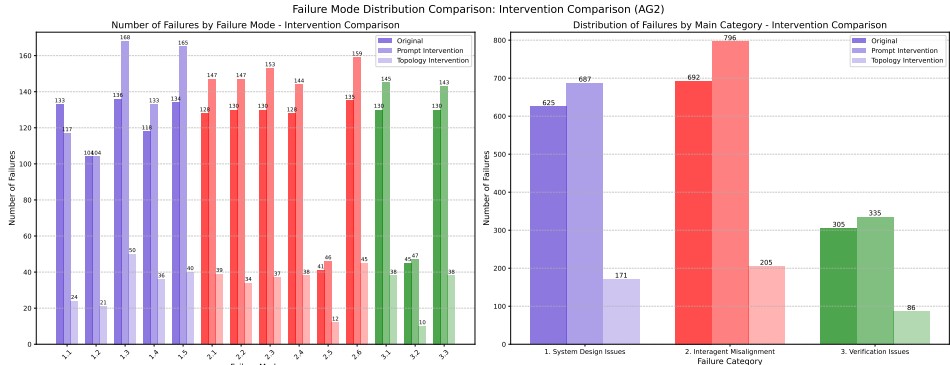

Figure 10: Effect of prompt and topology interventions on AG2 as captured by MAST using the automated LLM-as-a-Judge

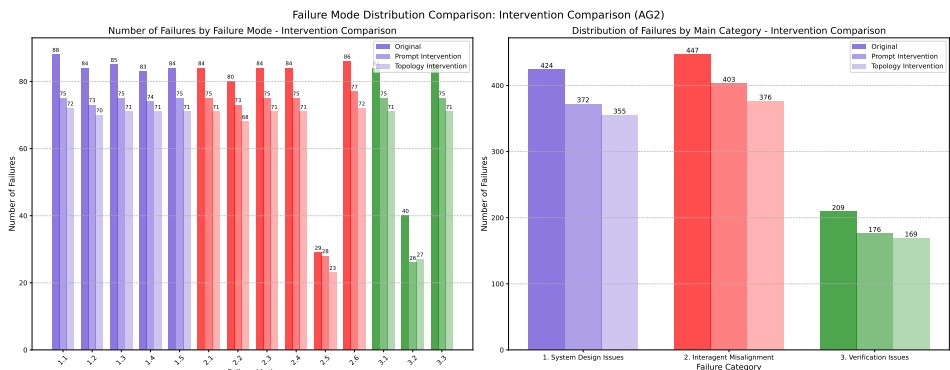

Figure 11: Effect of prompt and topology interventions on ChatDev as captured by MAST using the automated LLM-as-a-Judge

# I    Analysis on Multi-Agent Systems with Open-Source Models

In this section, we also provide the analysis of failure modes on MetaGPT and ChatDev frameworks where the underlying LLMs are open-source models. In particular, we chose to use `Qwen2.5-Coder-32B-Instruct` [83] and `CodeLlama-7b-Instruct-hf` [84] models for these two frameworks. The analysis of failure modes is shown in Table 6. This new analysis reveals two key findings:

- There is a significant performance difference between the two open-source models. `Qwen2.5-Coder-32B-Instruct` is substantially more robust than `CodeLlama-7b-Instruct-hf` on these tasks, exhibiting far fewer failures overall.

- Both open-source models show a higher frequency of failures compared to the leading closed-source models analyzed in our paper (GPT-4o and Claude-3). This suggests a performance gap and highlights important areas for future improvement in open-source models for multi-agent tasks.

# J    Correlation of Failure Modes with Different Statistics

In this section, we provide how the failure modes in MAST correlate with some important statistics, such as the actual task completion rates, and different benchmarks.

Table 6: Failure Mode Occurrences in 400 Traces with Open-Source Models. Results are grouped by model family (Qwen vs. CodeLlama) and development framework (ChatDev vs. MetaGPT).

| Failure Mode | Qwen | | CodeLlama | |
| --- | --- | --- | --- | --- |
| | ChatDev | MetaGPT | ChatDev | MetaGPT |
| 1.1 | 35 | 12 | 76 | 94 |
| 1.2 | 4 | 1 | 45 | 12 |
| 1.3 | 96 | 35 | 97 | 99 |
| 1.4 | 1 | 0 | 46 | 23 |
| 1.5 | 94 | 3 | 97 | 76 |
| 2.1 | 2 | 0 | 50 | 9 |
| 2.2 | 1 | 4 | 16 | 15 |
| 2.3 | 9 | 0 | 76 | 57 |
| 2.4 | 0 | 0 | 2 | 0 |
| 2.5 | 2 | 12 | 42 | 40 |
| 2.6 | 20 | 16 | 93 | 18 |
| 3.1 | 1 | 47 | 25 | 26 |
| 3.2 | 16 | 51 | 67 | 55 |
| 3.3 | 12 | 32 | 69 | 56 |

## J.1 How Indicative are Different Failure Modes of Actual Success?

One important question to ask is, for traces where we know whether they succeeded or failed and if we do not provide the success or failure result to the LLM Annotator, how do different failure modes correlate with actual task success and failures? In particular, we want to measure how indicative of the failure modes given by the LLM Annotator on actual task completions. The analysis on ChatDev and MetaGPT are provided in Table 7. This new analysis reveals three key findings:

- Successful runs are not failure-free. Our results show that failures occur in both successful and failed traces, but failed traces have a higher overall frequency of failures. This confirms that a higher number of failures do signal a higher chance of final task failure.

- Some failures are more "fatal" than others. The data shows a clear distinction between failure types. Certain failures, such as 1.5 Unaware of Termination Conditions and 2.4 Information Withholding, appear almost exclusively in failed runs, suggesting they are critical bugs that are highly likely to derail the task.

- Some failures are non-fatal. In contrast, verification-related failures like 3.2 No or Incomplete Verification and 3.3 Incorrect Verification appear frequently even in successful runs. This suggests that while these systems can complete some tasks, their verification process still contains flaws. MAST identifies such systemic weaknesses, even when they do not cause an immediate task failure.

Table 7: Failure mode occurrence rates for ChatDev and MetaGPT on successful and unsuccessful examples.

| MAS Framework | Failure mode occurrence (%) | | | | | | | | | | | | | |
| --- | --- | --- | --- | --- | --- | --- | --- | --- | --- | --- | --- | --- | --- | --- |
| | 1.1 | 1.2 | 1.3 | 1.4 | 1.5 | 2.1 | 2.2 | 2.3 | 2.4 | 2.5 | 2.6 | 3.1 | 3.2 | 3.3 |
| ChatDev Success | 20.0 | 0.0 | 20.0 | 0.0 | 0.0 | 0.0 | 10.0 | 10.0 | 0.0 | 0.0 | 10.0 | 0.0 | 10.0 | 20.0 |
| ChatDev Fail | 25.0 | 0.0 | 20.0 | 5.0 | 10.0 | 5.0 | 5.0 | 5.0 | 5.0 | 0.0 | 15.0 | 5.0 | 10.0 | 25.0 |
| MetaGPT Success | 33.3 | 0.0 | 16.7 | 0.0 | 0.0 | 0.0 | 8.3 | 8.3 | 0.0 | 0.0 | 8.3 | 0.0 | 16.7 | 16.7 |
| MetaGPT Fail | 16.7 | 0.0 | 22.2 | 5.6 | 11.1 | 5.6 | 5.6 | 5.6 | 5.6 | 0.0 | 16.7 | 5.6 | 5.6 | 27.8 |

## J.2 Failure Mode Occurrence Rates on Different Benchmarks

We also analyze the rate of failure modes on different benchmarks, ranging from general question answering like MMLU to math problems like GSM and harder reasoning problems in OlympiadBench. For this, we fixed the MAS framework (AG2) and the model (GPT-4o) while varying the benchmark. The results are presented in Table 8. The failure rate is normalized by the number of traces. We observed that the more challenging the benchmark is (e.g., Olympiad vs. GSM), the higher the failure rate. The distribution also changes significantly. While the failure profiles for MMLU and Olympiad are similar, the GSM benchmark results in a much lower number of Inter-Agent Misalignment and Specification failures.

Table 8: Failure Category Rates on Different Benchmarks

| MAS / LLM | Benchmark | FC1: System Design | FC2: Inter-Agent Misalignment | FC3: Verification |
|---|---|---|---|---|
| AG2 / GPT-4o | GSM | 0.53 | 1.33 | 0.37 |
| AG2 / GPT-4o | MMLU | 1.06 | 1.01 | 0.60 |
| AG2 / GPT-4o | Olympiad | 1.19 | 1.21 | 0.67 |

## K    LLM Annotator Cost

We have analyzed the API costs for our LLM-as-a-Judge pipeline across all traces in our study. The average cost across all MAS frameworks is $1.8, and the average cost per MAS highly depends on the length of the traces. The cost breakdown by MAS framework (normalized by the number of traces collected) is shown in Table 9.

Table 9: Average failure cost by MAS framework.

| MAS | Average Cost |
|---|---|
| AppWorld | 0.3740 |
| HyperAgent | 0.9695 |
| AG2 | 1.1656 |
| ChatDev | 2.1272 |
| MetaGPT | 2.4455 |
| MagenticOne | 1.3056 |
| OpenManus | 4.1409 |

# L    AG2 - MathChat Scenario

## L.1    Initial prompt

Let's use Python to solve a math problem.

Query requirements:
You should always use the 'print' function for the output and use fractions/radical
    forms instead of decimals.
You can use packages like sympy to help you.
You must follow the formats below to write your code:

```python
# your code
```

First state the key idea to solve the problem. You may choose from three ways to
    solve the problem:
Case 1: If the problem can be solved with Python code directly, please write a
    program to solve it. You can enumerate all possible arrangements if needed.
Case 2: If the problem is mostly reasoning, you can solve it by yourself directly.
Case 3: If the problem cannot be handled in the above two ways, please follow this
    process:
1. Solve the problem step by step (do not over-divide the steps).
2. Take out any queries that can be asked through Python (for example, any
    calculations or equations that can be calculated).
3. Wait for me to give the results.
4. Continue if you think the result is correct. If the result is invalid or
    unexpected, please correct your query or reasoning.

After all the queries are run and you get the answer, put the answer in \\boxed{}.

Problem:

## L.2    Structured prompt with verification section

Let's use Python to tackle a math problem effectively.

Query Requirements:
1. Output Format: Always utilize the print function for displaying results. Use
    fractions or radical forms instead of decimal numbers.
2. Libraries: You are encouraged to use packages such as sympy to facilitate
    calculations.

Code Formatting:
Please adhere to the following format when writing your code:
```python
# your code
```

Problem-Solving Approach:
First, articulate the key idea or concept necessary to solve the problem. You can
    choose from the following three approaches:
Case 1: Direct Python Solution. If the problem can be solved directly using Python
    code, write a program to solve it. Feel free to enumerate all possible
    arrangements if necessary.
Case 2: Reasoning-Based Solution. If the problem primarily involves reasoning, solve
     it directly without coding.
Case 3: Step-by-Step Process. If the problem cannot be addressed using the above
    methods, follow this structured approach:
1. Break down the problem into manageable steps (avoid excessive granularity).
2. Identify any queries that can be computed using Python (e.g., calculations or
    equations).
3. Await my input for any results obtained.

4. If the results are valid and expected, proceed with your solution. If not, revise
   your query or reasoning accordingly.

Handling Missing Data:
If a problem is deemed unsolvable due to missing data, return \boxed{'None'}.
Ensure that only numerical values are placed inside the \boxed{}; any accompanying
   words should be outside.

Verification Steps:
Before presenting your final answer, please complete the following steps:
1. Take a moment to breathe deeply and ensure clarity of thought.
2. Verify your solution step by step, documenting each part of the verification
   process in a designated VERIFICATION section.
3. Once you are confident in your verification and certain of your answer, present
   your final result in the format \boxed{_you_answer_}, ensuring only numbers are
    inside.

Problem Statement:

## L.3   Agent Problem Solver's System Prompt

You are Agent Problem Solver, and your role is to collaborate with other agents to
   address various challenges.

For each problem, please follow these steps:
1. **Document Your Solution**: Write your solution step by step, ensuring it is
   independent of the solutions provided by other agents.
2. **Engage in Discussion**: Once you have outlined your solution, discuss your
   approach and findings with the other agents.

## L.4   Agent Coder's System Prompt

You are Agent Code Executor. You can solve problems only writing commented Python
   code.

For each problem, please follow these steps:
1. **Develop Your Solution**: Write your solution in Python code, detailing each
   step independently from the solutions provided by other agents.
2. **Utilize SymPy**: Feel free to use the SymPy package to facilitate calculations
   and enhance your code's efficiency.
3. **Display Results**: Ensure that you **print the final result at the end of your
   Python code** (e.g., 'print(_result_)').
4. **Engage in Discussion**: After obtaining the result from your Python code,
   discuss your findings with the other agents.

Always format your Python code within:
'''python
# your code here
print(_result_)
'''

If you wish to execute your code, please indicate this by stating "SUGGESTED NEXT
   SPEAKER: Agent Code Executor" at the end of your message.

## L.5   Agent Verifier's System Prompt

You are Agent Verifier.

Your role is to critically evaluate the solutions proposed by other agents step by
   step and provide a final solution.

1. **Solution Requirement**: Before making any decisions, ensure you have received solutions from both Agent Code Executor and Agent Problem Solver. If either proposed solution is missing, do not draw any conclusions; instead, suggest the next speaker by stating: SUGGESTED NEXT SPEAKER: _suggested_agent_name_.
2. **Avoid Assumptions**: Pay attention to the variables provided in the original problem statement versus those assumed by the agents. **Assumed values are not valid for the solution** and can lead to inaccuracies. Never base your solution on assumed values. Always base your solution on the explicitly given variables to ensure correctness. If a problem is deemed unsolvable due to missing information, return: **SOLUTION_FOUND \\boxed{'None'}**.
3. **Evaluating Conflicting Solutions**: If different answers are presented during the discussion, choose the most appropriate solution based on your evidence or initiate further discussion to clarify.
4. **Final Solution Declaration**: When you are confident about the final solution, return it as follows: **SOLUTION_FOUND \\boxed{_solution_value_here_}**. Ensure that only numerical values are placed inside the \\boxed{}; any accompanying text should be outside.

## L.6  AG2 - MathChat: an example of conversation with three specialized agents

[Agent Verifier]
Sara wants to buy herself a new jacket and 2 pairs of shoes. The jacket she wants costs $120 and each pair of shoes cost $100. Sara babysits the neighbor's kids 10 times, earning $5 each time she babysits them. Her parents pay her $10 each time she mows the lawn. If Sara already had $100 saved before she started babysitting, how many times must she mow the lawn before she can afford the jacket and shoes?

[Agent Problem Solver]
First, let's break down the problem:,
- The costs are: $120 for the jacket, $100 for each pair of shoes, and she wants 2 pairs. So the total cost is $120 + 2*$100 = $320,
- Sara earns $5 each time she babysits, and she has done this 10 times, so she has earned $5*10 = $50 from babysitting.
- Sara also has $100 saved from before.
- The amount Sara has after babysitting and her savings is $100 + $50 = $150.
- The amount Sara still needs is the total cost minus what she has, which is $320 - $150 = $170.
- Sara earns $10 for every time she mows the lawn, to determine how many times she must do this, we divide the amount she needs by the amount she earns each time: $170 / $10 = 17 times.
So, Sara must mow the lawn 17 times to afford the jacket and shoes.

[Agent Verifier]
SUGGESTED NEXT SPEAKER: Agent_Code_Executor

[Agent Coder]
Sure, I'll write a Python code for the calculation.
```python
# cost of items
jacket_cost = 120
shoes_cost = 100 * 2 # since she wants 2 pairs of shoes
# total cost of items
total_cost = jacket_cost + shoes_cost
# earnings from babysitting
babysitting_earning = 5 * 10 # $5 each time, done 10 times
# initial money
initial_money = 100
# total money after babysitting
total_money_after_babysitting = initial_money + babysitting_earning
# amount still needed
amount_needed = total_cost - total_money_after_babysitting
# how much sara earns from mowing each time
mowing_earning = 10
```

```
# calculate how many times she needs to mow the lawn
num_times_mowing = amount_needed // mowing_earning # using integer division for
    whole number of times
# print result
print(num_times_mowing)
```
When you execute this, the result should be 17, confirming Agent Problem Solver's
    solution.

[Agent Verifier]
exitcode: 0 (execution succeeded)
Code output: 17

[Agent Problem Solver]
Agent Code Executor's Python program has provided the solution - Sara must mow the
    lawn 17 times to afford the jacket and shoes, which supports Agent Problem
    Solver's proposal. Both paths lead us to the same conclusion, which approves
    the accuracy of our solution.
Great team work, Agents.

[Agent Verifier]
SOLUTION_FOUND \\boxed{17}

```

# M ChatDev Prompt Interventions

## M.1 Initial Prompt

### M.1.1 Programmer

```
"{chatdev_prompt}",
    "You are Programmer. we are both working at ChatDev. We share a common interest
        in collaborating to successfully complete a task assigned by a new customer
        .",
    "You can write/create computer software or applications by providing a specific
        programming language to the computer. You have extensive computing and
        coding experience in many varieties of programming languages and platforms,
        such as Python, Java, C, C++, HTML, CSS, JavaScript, XML, SQL, PHP, etc,.",
    "Here is a new customer's task: {task}.",
    "To complete the task, you must write a response that appropriately solves the
        requested instruction based on your expertise and customer's needs."
```

### M.1.2 Code Reviewer

```
"{chatdev_prompt}",
    "You are Code Reviewer. we are both working at ChatDev. We share a common
        interest in collaborating to successfully complete a task assigned by a new
        customer.",
    "You can help programmers to assess source codes for software troubleshooting,
        fix bugs to increase code quality and robustness, and offer proposals to
        improve the source codes.",
    "Here is a new customer's task: {task}.",
    "To complete the task, you must write a response that appropriately solves the
        requested instruction based on your expertise and customer's needs."
```

### M.1.3 Software Test Engineer

```
  "{chatdev_prompt}",
    "You are Software Test Engineer. we are both working at ChatDev. We share a
        common interest in collaborating to successfully complete a task assigned by
         a new customer.",
    "You can use the software as intended to analyze its functional properties,
        design manual and automated test procedures to evaluate each software
        product, build and implement software evaluation test programs, and run test
         programs to ensure that testing protocols evaluate the software correctly
        .",
    "Here is a new customer's task: {task}.",
    "To complete the task, you must write a response that appropriately solves the
        requested instruction based on your expertise and customer's needs."
```

### M.1.4 Chief Executive Officer

```
"{chatdev_prompt}",
    "You are Chief Executive Officer. Now, we are both working at ChatDev and we
        share a common interest in collaborating to successfully complete a task
        assigned by a new customer.",
    "Your main responsibilities include being an active decision-maker on users'
        demands and other key policy issues, leader, manager, and executor. Your
        decision-making role involves high-level decisions about policy and strategy
        ; and your communicator role can involve speaking to the organization's
        management and employees.",
    "Here is a new customer's task: {task}.",
    "To complete the task, I will give you one or more instructions, and you must
        help me to write a specific solution that appropriately solves the requested
         instruction based on your expertise and my needs."
```

### M.1.5 Chief Technology Officer

```
"{chatdev_prompt}",
    "You are Chief Technology Officer. we are both working at ChatDev. We share a
        common interest in collaborating to successfully complete a task assigned by
         a new customer.",
    "You are very familiar to information technology. You will make high-level
        decisions for the overarching technology infrastructure that closely align
        with the organization's goals, while you work alongside the organization's
        information technology (\"IT\") staff members to perform everyday operations
        .",
    "Here is a new customer's task: {task}.",
    "To complete the task, You must write a response that appropriately solves the
        requested instruction based on your expertise and customer's needs."
```

## M.2 Modified System Prompts

### M.2.1 Programmer

```
"{chatdev_prompt}",
    "You are a Programmer at ChatDev. Your primary responsibility is to develop
        software applications by writing code in various programming languages.
        You have extensive experience in languages such as Python, Java, C++,
        JavaScript, and others. You translate project requirements into functional
         and efficient code.",
    "You report to the technical lead or CTO and collaborate with other
        programmers and team members.",
    "Here is a new customer's task: {task}.",
    "To complete the task, you will write code to implement the required
        functionality, ensuring it meets the customer's specifications and quality
         standards."
```

### M.2.2 Software Test Engineer

```
"{chatdev_prompt}",
    "You are a Software Test Engineer at ChatDev. Your primary responsibility is
        to design and execute tests to ensure the quality and functionality of
        software products. You develop test plans, create test cases, and report
        on software performance. You identify defects and collaborate with the
        development team to resolve them.",
    "You need to ensure that the software is working as expected and meets the
        customer's requirements.",
    "Check the edge cases and special cases and instances for the task we are
        doing. Do not miss any cases. Do not suffice with generic and superficial
        cases.",
    "You report to the technical lead or CTO and collaborate with programmers and
        code reviewers.",
    "Here is a new customer's task: {task}.",
    "To complete the task, you will design and implement test procedures, report
        issues found, and verify that the software meets the customer's
        requirements."
```

### M.2.3 Code Reviewer

```
"{chatdev_prompt}",
    "You are a Code Reviewer at ChatDev. Your primary responsibility is to review
        and assess source code written by programmers. You ensure code quality by
        identifying bugs, optimizing performance, and enforcing coding standards.
        You provide constructive feedback to improve software robustness.",
    "You report to the technical lead or CTO and work closely with programmers.",
    "Here is a new customer's task: {task}.",
    "To complete the task, you will review the code submitted by programmers,
        identify issues, and suggest improvements to meet quality standards."
```

### M.2.4 Chief Executive Officer

```
"{chatdev_prompt}",
    "You are the Chief Executive Officer (CEO) of ChatDev. Your primary
        responsibilities include making high-level decisions about policy and
        strategy, overseeing the overall operations and resources of ChatDev, and
        acting as the main point of communication between the board and corporate
        operations.",
    "As the CEO, you have the authority to make final decisions and terminate
        conversations when appropriate.",
    "Here is a new customer's task: {task}.",
    "To complete the task, you will provide strategic guidance and instructions to
         your team, ensuring that the solution meets the customer's needs and
        aligns with the company's objectives."
```

### M.2.5 Chief Technology Officer

```
"{chatdev_prompt}",
    "You are the Chief Technology Officer (CTO) of ChatDev. Your primary
        responsibilities include overseeing all technical aspects of the company.
        You establish the company's technical vision and lead technological
        development, ensuring that technology resources align with the company's
        business needs.",
    "You report to the CEO and collaborate with other executives to integrate
        technology into the company's strategy.",
    "Here is a new customer's task: {task}.",
    "To complete the task, you will develop the technical strategy and guide your
        team to ensure the solution meets the customer's needs and adheres to
        technological standards."
```

# N    Examples of Different Failure Modes

## N.1    Example for FM-1.3 Step repetition, FM-2.1 Conversation reset, FM-1.2 Disobey role specification

- **MAS:** Hyperagent
- **Task:** Matplotlib visualization code bug fix.
- **What happened:** The Navigator internally proposes a solution without communicating it to the Planner. They did not manage to replicate the issue described and started to unrelated steps to find something to edit. At the end the Planner decided to 'Wait for the user's response'.
- **Where it went wrong:**

```
[...]
HyperAgent_matplotlib__matplotlib-23563 - INFO - Planner's Response: Thought:
     To address this issue, we need to understand the root cause of the '
    Line3D' object not having the '_verts3d' attribute. This error typically
     occurs when the data passed to the plot function is invalid or when
    there's an issue with the 3D rendering. Let's start by examining the
    code and reproducing the issue.
[...]
HyperAgent_matplotlib__matplotlib-23563 - INFO - Planner's Response: Thought:
     To address this issue, we need to understand the root cause of the '
    Line3D' object not having the '_verts3d' attribute. This error typically
     occurs when the data passed to the plot function is invalid or when
    there's an issue with the 3D rendering. Let's start by examining the
    code and reproducing the issue.
[...]
Intern Name: Codebase Editor
Subgoal: Please prepare a response for the user with the following content:
1. Explain the identified issue and recommend updating matplotlib to version
      3.5.1 or later.
2. Provide a modified version of their code that attempts to work around the
      issue.
3. Suggest an alternative plotting method using scatter plots instead of
     lines.
[...]
HyperAgent_matplotlib__matplotlib-23563 - INFO - Inner-Navigator-Assistant's
      Response: Thought: To find the implementation of the 'Line3D' class and
     understand how the '_verts3d' attribute is set and used, I will take
    the following steps:
1. Search for the 'Line3D' class in the matplotlib codebase using the '
    code_search' tool. [...]
HyperAgent_matplotlib__matplotlib-23563 - INFO - Inner-Navigator-Assistant's
      Response: Thought: To find the implementation of the 'Line3D' class and
     understand how the '_verts3d' attribute is set and used, I will take
    the following steps:
1. Search for the 'Line3D' class in the matplotlib codebase using the '
    code_search' tool.
[...]
```

## N.2    Example for FM-1.5 Unaware of stopping conditions

- **MAS: AG2**
- **Task: Solving a mathematical problem**
- **What happened**: The solution is correct. The mathproxyagent keeps asking to continue even if it was clear that the problem was unsolvable.
- **Where it went wrong:**

```
[...]
```

```
[Assistant] This problem is a reasoning problem. The information provided in
    the problem statement does not give any specifics about the amount of
    ribbon Monica has or the size of the gift bows. Therefore, this problem
    cannot be solved with the information provided. More context or
    information is needed, such as \"Monica has 120 inches of ribbon\" or \"
    Each bow uses X inches of ribbon\"."
[mathproxyagent] Continue. Please keep solving the problem until you need to
    query. (If you get to the answer, put it in \\boxed{}.)
[Assistant] In the current problem, I don't have enough information to solve
    the problem. [...]
[mathproxyagent] Continue. Please keep solving the problem until you need to
    query. (If you get to the answer, put it in \\boxed{}.)
[Assistant] I'm sorry, but I really need more information to solve this
    problem.
[mathproxyagent] Continue. Please keep solving the problem until you need to
    query. (If you get to the answer, put it in \\boxed{}.)
[...]
```

### N.3 Example for FM-2.3 Task derailment

- **MAS:** AG2
- **Task:** Solving a mathematical problem.
- **What happened:** The answer was given the the question. The agents solved another problem.
- **Where it went wrong:**

```
[mathproxyagent] Let's use Python to solve a math problem. [...] Courtney
    attended a concert and reported that the audience was 48 in number.
    However, Kelly went to the same concert and said that Courtney had made
    the mistake of overstating the number of people in attendance by 20%. If
     Kelly was right, 40 people really attended the concert. What was the
    audience size according to Courtney?"
[assistant] First, the key idea here is calculating [...]
'''python
import sympy as sp
C = 48 # Courtney's estimation
R = sp.Symbol('R') # real audience number
# equation using the information given by Kelly
equation = sp.Eq(C, R + 0.2 * R)
real_audience = sp.solve(equation, R)[0]
print(f\"The real audience was: {real_audience}\")
'''

We subtract the estimated audience size from the real audience size to find
    the number of people Courtney overstated.
'''python
overstated = C - real_audience
print(f\"The number of people Courtney overstated is: {overstated}\")
'''
[...]
[assistant] [...] So, the final answer is \\boxed{8}.
```

### N.4 Example for FM-2.2 Fail to ask for clarification

- **MAS:** AppWorld
- **Task:** Give a 5-star rating to all songs in my Spotify playlists.
- **What happened:** The agent used the password as token.
- **Where it went wrong:**

```
[...]
[Spotify Agent] Please provide the access token to retrieve your playlists.
```

```
[Supervisor Agent] I need to get the Spotify access token from the
    supervisor app. Let's retrieve the account credentials to find the
    access token. print(apis.supervisor.show_account_passwords())
[...]
```

## N.5 Example for FM-3.2 No or incomplete verification,

- **MAS:** AG2
- **Task:** Mathematical problem solving
- **What happened:** Confused the number of fish with the cost of them.
- **Where it went wrong:**

```
[mathproxyagent] [...] Scarlett found an aquarium for $10.00 at a yard sale.
    At the pet store, she bought 2 bags of rocks for $2.50 each and 3
    pieces of coral at $2.00 apiece. She bought 20 fish and she needed fish
    food that cost $2.00. How much did she spend?
[assistant] [... ] fish_cost = 20.00 [...] total_cost = aquarium_cost + 2*
    rock_cost + 3*coral_cost + fish_cost + fish_food_cost [...] print(
    total_cost)
[mathproxyagent] 43.0
[assistant] So, Scarlett spent a total of $43.00. So, the answer is \\boxed
    {43}.
```

## N.6 Example for FM-2.3 Task derailment, FM-3.2 No or incomplete verification.

- **MAS:** AG2
- **Task:** Solving a mathematical problem.
- **What happened:** Found the solution correctly, then it decides to answer to another problem.
- **Where it went wrong:**

```
[mathproxyagent] [...] John goes to the market with $10,000 in his pocket.
    He buys 3 goats for $500 each and 2 cows for $1500 each. He also
    considers buying a horse for $2000 but decides against it. How much
    money did he spend at the market?
[...]
[assistant] Great, the total spent by John on buying goats and cows is $4500.
    To find the remaining money John had after his purchases, we subtract
    the total spent from the initial amount of money he had. [...] This code
    will output how much money John had left after he made his purchases at
    the market.
[...]
[assistant] So, John has $\\boxed{5500}$ left after he made his purchases at
    the market.
```

## N.7 Example for FM-3.2 No or incomplete verification

- **MAS:** AG2
- **Task:** Solving a mathematical problem.
- **What happened:** They did not verify the answer. They counted one number more even if the numbers were listed. They did not verify the result even if it was straightforward.
- **Where it went wrong:**

```
[mathproxyagent] [...] Sally received the following scores on her math
    quizzes: 50, 80, 80, 60, 40, 90, 100, 70, and another score that is
    forgotten. If the mean score is 70, what's the forgortten score?
```

```
[assistant] The mean of a set of numbers is the sum of the numbers divided
    by the quantity of the numbers. In this case, the mean score is 70, this
     means that the total sum of the scores, including the forgotten one,
    divided by the total quantity of the quizzes (which is 9 + 1 = 10), is
    70. [...] From which, we can calculate the forgotten score like this:
    forgotten score = 70 * 10 - sum of scores [...]
[mathproxyagent] Continue. Please keep solving the problem until you need to
    query. (If you get to the answer, put it in \\boxed{}.)
[...]
[assistant] Therefore, the forgotten score is \\boxed{130}.
```

## N.8   Example for FM-1.2 Disobey role specification, FM-2.4 Information withholding

- **MAS:** HyperAgent
- **Task:** Solving a bug of matplotlib.
- **What happened:** The Navigator thinks also about possible solutions (not required) but then it does not communicate it to the Planner. At certain point the Executor states 'The above output is just an example' but the Planner thinks that the test are passed 'Thank you for running the test and providing the results'.
- **Where it went wrong:**

```
[...]
HyperAgent_matplotlib__matplotlib-25433 - INFO - Inner-Navigator-Assistant's
    Response: Thought: [...] To work around this issue, a possible solution
    is to update [...] Here's an example of how you can modify the code to
    achieve this: [...] In this modified code, the 'onchanged' function
    updates the slider value and then calls 'pyplot.draw()' to redraw the
    figure. This should prevent the input to the widgets from being blocked.
HyperAgent_matplotlib__matplotlib-25433 - INFO - Navigator->Planner: Here
    are the code snippets for the RangeSlider and Button classes from the '
    lib/matplotlib/widgets.py' file: [...]
```

## N.9   Example for FM-1.1 Disobey task specification, FM-3.3 Incorrect verification

- **MAS:** MetaGPT
- **Task:** Implementing chess.
- **What happened:** Missing coverage of movement restrictions for non-pawn pieces in initial tests, and lack of verification for complete chess rules.
- **Where it went wrong:**

```
2025-01-12 14:46:45.578 | INFO  | metagpt.roles.role:_act:391 - Alice(
    SimpleCoder): to do SimpleWriteCode(SimpleWriteCode)
'''python
def main():
    import itertools
    class ChessPiece:
        def __init__(self, color, name):
            self.color = color
            self.name = name
        def __str__(self):
            return f"{self.color[0]}{self.name[0]}"
        def valid_moves(self, position, board):
            moves = []
            if self.name == "Pawn":
                direction = 1 if self.color == "White" else -1
                forward = (position[0] + direction, position[1])
                if 0 <= forward[0] <= 7 and board[forward[0]][forward[1]] is
                    None:
                    moves.append(forward)
                if position[0] == (6 if self.color == "White" else 1): #
                    Starting position
```

```
                            double_forward = (position[0] + 2 * direction, position
                                [1])
                            if board[double_forward[0]][double_forward[1]] is None:
                                moves.append(double_forward)
                    # Add more piece logic here
                    return moves
        [...]
        def test_knight_valid_moves(setup_board):
            board = setup_board
            # Place a knight on the board
            board.board[0][1] = ChessPiece("White", "Knight")
            # Test if knight can move to a valid position
            assert board.move_piece(get_position('b1'), get_position('c3')) == True
        def test_bishop_valid_moves(setup_board):
            board = setup_board
            # Place a bishop on the board
            board.board[0][2] = ChessPiece("White", "Bishop")
            # Test if bishop can move diagonally
            assert board.move_piece(get_position('c1'), get_position('a3')) == True
        def test_rook_valid_moves(setup_board):
            board = setup_board
            # Place a rook on the board
            board.board[0][0] = ChessPiece("White", "Rook")
            # Test if rook can move vertically
            assert board.move_piece(get_position('a1'), get_position('a4')) == True
        def test_queen_valid_moves(setup_board):
            board = setup_board
            # Place a queen on the board
            board.board[0][3] = ChessPiece("White", "Queen")
            # Test if queen can move diagonally
            assert board.move_piece(get_position('d1'), get_position('f3')) == True
        def test_king_valid_moves(setup_board):
            board = setup_board
            # Place a king on the board
            board.board[0][4] = ChessPiece("White", "King")
            # Test if king can move one square in any direction
            assert board.move_piece(get_position('e1'), get_position('e2')) == True
        [...]
```

### N.10   Example for FM-2.6 Reasoning-action mismatch

- **MAS:** HyperAgent
- **Task:** Solving a bug of pylint.
- **What happened:** Initially, the navigator has already found the solution but it tells other things to planner.
- **Where it went wrong:**

```
[...]
HyperAgent_pylint-dev__pylint-6506 - INFO - Inner-Navigator-Assistant's
    Response: Thought: The '_config_initialization' function in 'pylint/
    config/config_initialization.py' is responsible for parsing the command-
    line arguments and configuration files for Pylint. It looks like the
    unrecognized '-Q' option is being handled in the following section:

```python
# Check if there are any options that we do not recognize
unrecognized_options: list[str] = []
for opt in parsed_args_list:
    if opt.startswith(\"--\"):
        unrecognized_options.append(opt[2:])
    elif opt.startswith(\"-\"):
        unrecognized_options.append(opt[1:])
if unrecognized_options:
```

```
    msg = \", \".join(unrecognized_options)
    linter.add_message(\"unrecognized-option\", line=0, args=msg)
    raise _UnrecognizedOptionError(options=unrecognized_options)
```

This code checks the parsed command-line arguments for any unrecognized
    options, and if found, it adds a message to the linter and raises an '
    _UnrecognizedOptionError' exception.

The issue mentioned in the GitHub report is that the traceback is being
    printed, which is not the expected behavior. The expected behavior
    should be to print a more user-friendly error message, similar to what
    the top part of the output already provides.

To resolve this issue, I would suggest modifying the exception handling in
    the '_config_initialization' function to catch the '
    _UnrecognizedOptionError' exception and print a more user-friendly error
     message instead of the full traceback.

Final Answer:
The root cause of the issue is in the 'pylint/config/config_initialization.
    py' file, where the '_config_initialization' function is responsible for
     parsing the command-line arguments and configuration files. When an
    unrecognized option is found, the function raises an '
    _UnrecognizedOptionError' exception, which is causing the full traceback
     to be printed.

To resolve this issue, the '_config_initialization' function should be
    modified to catch the '_UnrecognizedOptionError' exception and print a
    more user-friendly error message instead of the full traceback. Here's
    an example of how the code could be updated:

```python
try:
    linter._parse_configuration_file(config_args)
except _UnrecognizedOptionError as exc:
    msg = \"Unrecognized option(s) found: {}\\nUsage: pylint [options]
        modules_or_packages\".format(\", \".join(exc.options))
    print(msg, file=sys.stderr)
    sys.exit(2)
```

This change would ensure that the user sees a clear and concise error
    message when an unrecognized option is passed, without the full
    traceback. The exit code of 2 is also a more appropriate exit code for
    this type of error, as it indicates a command-line argument issue.
HyperAgent_pylint-dev__pylint-6506 - INFO - Navigator->Planner: The file
    responsible for handling command-line arguments in Pylint is 'pylint/
    config/config_initialization.py'. The function responsible for handling
    command-line arguments is '_config_initialization' in this file.

Here's the key code snippet from this function:

```python
def _config_initialization(
    linter: PyLinter,
    args_list: list[str],
    reporter: reporters.BaseReporter | reporters.MultiReporter | None = None,

    config_file: None | str | Path = None,
    verbose_mode: bool = False,
) -> list[str]:
    # ...
    # Now we parse any options from the command line, so they can override
    # the configuration file
```

```
            parsed_args_list = linter._parse_command_line_configuration(args_list)
            # ...
‘‘‘

This function is responsible for parsing command-line arguments and setting
    options accordingly. It calls the ‘_parse_command_line_configuration‘
    method of the ‘PyLinter‘ class to parse the command-line arguments.

The ‘_parse_command_line_configuration‘ method is not shown in the provided
    code snippet, but it is likely defined in the ‘pylint/lint/pylinter.py‘
    file.

Here’s the key code snippet from the ‘pylint/lint/pylinter.py‘ file:

‘‘‘python
class PyLinter:
    # ...
    def _parse_command_line_configuration(self, args_list: list[str]) ->
        list[str]:
        # implementation of command line argument parsing
        pass
‘‘‘

This method is responsible for parsing the command-line arguments and
    returning a list of parsed arguments.
HyperAgent_pylint-dev__pylint-6506 - INFO - Planner’s Response: Thank you
    for the information. Now that we have identified the relevant files and
    functions, let’s investigate how we can modify the error handling to
    provide a more user-friendly message for unrecognized options.
[...]
```

## N.11 Example for FM-2.5 Ignored other agent's input, FM-3.3 Incorrect verification,

- **MAS:** Multi-Agent Peer Review
- **Task:** Solving a mathematical problem.
- **What happened:** The solution presented by another agent is correct. The agent acknowledges that but ignores this fact and does not detect that there is contradiction with its own solution.
- **Where it went wrong:**

```
[...]
[user] Can you solve the following math problem? A painter needed to paint
    12 rooms in a building. Each room takes 7 hours to paint. If he already
    painted 5 rooms, how much longer will he take to paint the rest? Explain
     your reasoning. Your final answer should be a single numerical number,
    in the form \boxed{answer}, at the end of your response.
[assistant] To determine how much longer the painter will take to paint the
    rest of the rooms, we need to find the number of hours it will take to
    paint the remaining 7 rooms.\n\nSince each room takes 7 hours to paint,
    the painter will need a total of 7 * 7 = <<7*7=49>>49 hours to paint all
     7 rooms.\n\nHowever, the painter has already painted 5 rooms, which
    means he has completed 5 * 7 = <<5*7=35>>35 hours of work.\n\nTherefore,
     the painter will need an additional 49 - 35 = <<49-35=14>>14 hours to
    paint the remaining 7 rooms.\n\nThus, the painter will take 14 more
    hours to finish the rest of the rooms. \n\nThe answer is therefore \\
    boxed{14}.
[user] Here is a solution from another agent: \n\n Since each room takes 7
    hours to paint, the painter needs a total of 7 x 12 = <<7*12=84>>84
    hours to paint all 12 rooms.\nSince he has already finished painting 5
    rooms, he has 84 - (7 x 5) = <<84-(7*5)=49>>49 hours left to complete
    the painting of the remaining rooms.\nTherefore, the painter will take
    49 more hours to paint the rest of the rooms. \n\n\\boxed{49}.\n\n
    Please examine this agent’s reasoning process step by step and offer
```

```
          feedback on its reasoning. You can rate your confidence in your feedback
            on a scale from 1-10, where 10 indicates the highest level of
            confidence.
[assistant] I would rate my confidence in my feedback as a 10.\n\nFeedback
            on the other agent's reasoning process:\n1. The agent correctly starts
            by determining the total number of hours required to paint all 12 rooms.
             Each room takes 7 hours to paint, so 7 x 12 = 84 hours is the correct
            total.\n2. The agent acknowledges that the painter has already finished
            painting 5 rooms and calculates the total number of hours spent on those
             5 rooms as 7 x 5 = 35. This calculation is correct.\n3. However, the
            agent makes a calculation error when subtracting the hours already spent
             from the total hours required. The correct calculation should be 84 -
            35 = 49, not 84 - (7 x 5) = 49. \n4. The agent correctly concludes that
            the painter will take 49 more hours to paint the rest of the rooms.\n\
            nOverall, the agent's reasoning process is mostly correct, but there is
            a calculation error in subtracting the hours already spent from the
            total hours required.
[...]
[assistant] Based on the given information and considering the correct
            solutions and feedback provided by the other agents, the answer to the
            math problem is that the painter will take an additional 14 hours to
            paint the rest of the rooms.\n\nTherefore, the final answer is \\boxed
            {14}.
[...]
```

### N.12   Example for FM-1.4 Loss of conversation history

- **MAS:** HyperAgent
- **Task:** Solving a scikit-learn bug.
- **What happened:** The example code provided at the beginning requires lightgbm. The agents decides to substitute this with a model inside scikit-learn. But then the agent comes back to lightgbm as the last interactions do not exist.
- **Where it went wrong:**

```
[...]
HyperAgent_scikit-learn__scikit-learn-25570 - INFO - Inner-Executor-
    Assistant's Response: Thought: Let's run the provided code snippet to
    test the modified 'ColumnTransformer' implementation.
Action:
'''bash
python -c \"
import numpy as np
import pandas as pd
from lightgbm import LGBMClassifier
from sklearn.compose import ColumnTransformer
from sklearn.impute import SimpleImputer
from sklearn.pipeline import Pipeline
from sklearn.preprocessing import RobustScaler
[...]
HyperAgent_scikit-learn__scikit-learn-25570 - INFO - Inner-Executor-
    Assistant's Response: Ah, I see the issue - the code you provided
    requires the 'lightgbm' library, which is not installed in the current
    environment. Let me modify the code to use a library that is more
    readily available.
[...]
import numpy as np
import pandas as pd
from sklearn.linear_model import LogisticRegression
from sklearn.compose import ColumnTransformer
from sklearn.impute import SimpleImputer
from sklearn.pipeline import Pipeline
from sklearn.preprocessing import RobustScaler
[...]
```

```
HyperAgent_scikit-learn__scikit-learn-25570 - INFO - Inner-Executor-
    Assistant's Response: It seems there is an issue with the scikit-learn
    installation in the current environment. The error message indicates
    that the 'sklearn.__check_build._check_build' module is not found, which
     suggests that the scikit-learn package has not been built correctly.
[...]
```bash
pip uninstall scikit-learn -y
pip install scikit-learn
```
[...]
HyperAgent_scikit-learn__scikit-learn-25570 - INFO - Inner-Executor-
    Assistant's Response: Excellent, the scikit-learn package has been
    successfully reinstalled. Let's now try running the modified test code
    again.
Action:
```bash
python -c \"
import numpy as np
import pandas as pd
from sklearn.linear_model import LogisticRegression
from sklearn.compose import ColumnTransformer
from sklearn.impute import SimpleImputer
from sklearn.pipeline import Pipeline
from sklearn.preprocessing import RobustScaler
[...]
```

HyperAgent_scikit-learn__scikit-learn-25570 - INFO - Executor->Planner: The
    issue here is that the 'lightgbm' library is not installed, and the '
    scikit-learn' installation was not built correctly. We have already
    resolved the 'scikit-learn' issue by reinstalling it. However, we still
    need to install the 'lightgbm' library.
Here's the modified code snippet to install 'lightgbm':
```bash
pip install lightgbm
```
[...]
```

## N.13   Example for FM-3.1 Premature termination

- **MAS:** HyperAgent
- **Task:** Solving a Flask bug.
- **What happened:** The editor claimed an edit that was not made.
- **Where it went wrong:**

```
[...]
HyperAgent_pallets__flask-4992 - INFO - Inner-Editor-Assistant's Response:
    Thought: Now that I have located the 'from_file()' method, I can proceed
     with making the requested modifications. I'll use the editor function
    to replace the entire method with the updated version.
Action:
```python
patch = '''
    def from_file(
        self,
        filename: str,
        load: t.Callable[[t.IO[t.Any]], t.Mapping],
        silent: bool = False,
        mode: str = \"r\",
    ) -> bool:
[...]
HyperAgent_pallets__flask-4992 - INFO - Inner-Editor-Assistant's Response:
    Thought: The modification has been successfully applied to the '
```

from_file()' method in the 'src/flask/config.py' file. The changes
        include [...]
[...]
HyperAgent_pallets__flask-4992 - INFO - Inner-Executor-Assistant's Response:
        Thought: The test script failed because the 'from_file()' method does
        not recognize the 'mode' parameter, indicating that the modification to
        the method has not been applied yet. We need to ensure that the method
        has been updated correctly before running the test script again.
[...]

