# OpenReview forum: "Why Do Multi-Agent LLM Systems Fail?"
_NeurIPS.cc/2025/Datasets_and_Benchmarks_Track — NeurIPS 2025 Datasets and Benchmarks Track spotlight_

### Official Review · Reviewer_4gU4 · 2025-06-10

**Rating:** 5
**Confidence:** 4

**Summary:**

This paper presents a systematic investigation into why Multi-Agent LLM Systems fail. The authors introduce two primary contributions: MAST, an empirically grounded taxonomy of 14 failure modes organized into 3 categories (Specification Issues, Inter-Agent Misalignment, and Task Verification), and MAD, a large-scale dataset of over 1,000 annotated MAS execution traces from 7 different frameworks. The MAST taxonomy was developed using Grounded Theory and validated with high inter-annotator agreement. To build the MAD dataset at scale, the authors developed and calibrated an LLM-as-a-Judge pipeline. The paper uses these resources to analyze failure distributions across different models and architectures and conducts intervention case studies to demonstrate MAST's practical utility as a diagnostic tool for improving MAS reliability.

**Additional Feedback:**

NA

**Dataset Code Accessibility:**

Yes

**Dataset Code Comments:**

https://huggingface.co/datasets/mcemri/MAD

**Ethical Considerations:**

No, there are no or only very minor ethics concerns

**Final Justification:**

After reading the rebuttal, I keep my score as it is.

**Limitations Weaknesses:**

- The paper posits that MAST helps identify the root causes of failures. However, the taxonomy sometimes seems to categorize symptoms whose root causes could be multifaceted.

**Strengths Contributions:**

- The paper addresses a critical and timely question in the burgeoning field of MAS. By moving beyond simple task success rates to ask why these systems fail, it provides a much-needed framework for a more principled and systematic approach to MAS development. The work is foundational and likely to be highly cited.
- The creation of the MAST taxonomy is methodologically sound.
- The release of the MAD dataset, the MAST taxonomy, and the LLM annotator is a significant contribution to the research community.

---

> ### Author Rebuttal · Authors · 2025-07-31
>
> Hi reviewer 4gU4,
>
> Thank you for your valuable time and strong scores. We really appreciate and are honoured by your high recognition of our work as "foundational and likely to be highly cited", and acknowledgement on the strength of our paper as:
> * **A critical and timely problem** in the burgeoning field of MAS.
> * It provides **a much-needed framework** for a more principled and **systematic approach** to MAS development.
> * The creation of the MAST taxonomy is **methodologically sound**.
> * The release of the MAD dataset, the MAST taxonomy, and the LLM annotator is a **significant contribution to the research community**.
>
> ***
> ## Failures vs. Root Causes:
>
> > "The paper posits that MAST helps identify the root causes of failures. However, the taxonomy sometimes seems to categorize symptoms whose root causes could be multifaceted."
>
> > **Action**: We will update the wording in the paper to clarify this distinction.
>
> Thank you for raising this important distinction. MAST, by design, focuses on systematically identifying and categorizing observable failure modes, not on performing a complete, automated root cause analysis (the *how* a failure happen).
>
> A useful analogy is a segmentation fault: MAST identifies the "segfault" (the observable failure), while a debugger would be needed to find the specific root cause (e.g., a null pointer dereference). Our contribution is to provide the foundational framework that systematically captures these observable failures to guide the subsequent, deeper investigation required for true root cause analysis. We will update the writing in the paper to cover the difference.
>
> ***
> ## Additional Enhancements to the Paper
> To further strengthen the paper, we also wanted to briefly share several significant additions made in response to feedback from the review process:
> 1. **Expanded Analysis on Open-Source Models**: We have generated and analyzed 400 new traces from powerful open-source models (`Qwen` and `CodeLlama`) to broaden the paper's findings and increase the value of our MAD dataset.
> 2. **Deeper Failure Analysis**: We now include new analyses breaking down failure distributions by benchmark and comparing failure patterns in
>  successful versus failed runs.
> 3. **New Open-Source Tools**: To maximize the practical impact of our work, we are releasing an open-source Python library and an interactive dashboard to make it easier for other researchers to apply our failure analysis pipeline to their own systems .
> ***
> Thank you again for your valuable time, strong score, and useful feedback for helping us improve the manuscript. We believe that with these clarifications and substantial new additions, further strengthen the contribution of our work to the community.
>
> Best,
>
> The Authors

---

### Official Review · Reviewer_yBzE · 2025-06-29

**Rating:** 5
**Confidence:** 4

**Summary:**

This paper presents a framework for analyzing failures in Multi-Agent LLM Systems (MAS) to understand why they often underperform. It introduces two primary contributions: the Multi-Agent System Failure Taxonomy (MAST), an empirically-derived classification of 14 failure modes, and the Multi-Agent system Failure Dataset (MAD), a large-scale dataset. MAST was developed through rigorous, data-driven analysis and validated with high inter-annotator agreement. To build the large-scale MAD dataset, the authors created a scalable LLM-as-a-Judge pipeline. The work demonstrates how these tools can guide targeted interventions, arguing that many failures stem from system design rather than just the underlying models.

**Dataset Code Accessibility:**

Partly

**Dataset Code Comments:**

The paper states that 21 instances annotated by human experts were released in the MAD-human dataset; however, we only found 19 at the provided open-source URL. Why did the author claim in the paper to have collected 150 instances and open-sourced 21, when only 19 were actually released? Could you please clarify this discrepancy?

**Ethical Considerations:**

No, there are no or only very minor ethics concerns

**Final Justification:**

This work opens an interesting direction towards evaluating and improving the performance of Multi-Agent LLM Systems. All of my concerns have been addressed by the rebuttal.

**Limitations Weaknesses:**

1.The MAST taxonomy, while thorough, may not be exhaustive. As the MAS landscape rapidly evolves, new architectures could introduce failure modes not yet captured.

2.The large-scale annotation depends on an LLM annotator. While its agreement with human experts is high (κ=0.77), it is not high enough, which introduces the possibility of some systemic bias in the final dataset's failure distributions.

3.MAST provides an excellent classification of observable failure modes. However, attributing these modes to a single root cause is inherently challenging in complex systems. A single failure mode like "FM 1.1: Disobey Task Specification" could arise from multiple potential causes, such as a poorly written user prompt, a flaw in the agent's role definition within the MAS, or an intrinsic limitation of the base LLM. While the taxonomy is a crucial first step, the path from classifying a failure to diagnosing its ultimate root cause remains a non-trivial challenge.

4.The analysis focuses primarily on leading closed-source models, which makes me wonder if expanding the work to powerful open-source models might reveal different failure patterns and broaden the findings. For example, the HyperAgent paper also conducted experiments using open-source models.

5.The paper states that 21 instances annotated by human experts were released in the MAD-human dataset; however, we only found 19 at the provided open-source URL. Why did the author claim in the paper to have collected 150 instances and open-sourced 21, when only 19 were actually released?Could you please clarify this discrepancy?

**Strengths Contributions:**

1.The paper addresses a critical and timely problem. As the community rapidly explores MAS for complex tasks like software engineering and scientific discovery, a systematic understanding of their failure points is essential for moving beyond hype and achieving reliable performance. This work directly tackles the observed gap between the performance of complex MAS and simpler baselines.

2.The paper contributes MAST and MAD, the first standardized taxonomy and large-scale annotated dataset for MAS failures. These are invaluable resources that will undoubtedly fuel future research.

3.The development and validation of the LLM annotator  makes the MAST framework highly practical and scalable for future use by the broader community.

4.The use of Grounded Theory and iterative Inter-Annotator Agreement (IAA) studies to create the MAST taxonomy is rigorous, ensuring the failure modes are empirically grounded and robustly defined.

---

> ### Author Rebuttal · Authors · 2025-07-31
>
> Hi Reviewer yBzE,
>
> Thank you for your valuable time and thorough review. We are grateful for your positive assessment and acknowledgment on:
> * A *critical and timely* problem in a rapidly evolving field.
> * MAST and MAD are *"invaluable resources"* that will *"undoubtedly fuel future research."*
> * Our methodology, including the use of Grounded Theory, is *rigorous and empirically grounded*.
> * The validated LLM annotator makes our framework *highly practical and scalable* for the community.
>
> Your feedback has been very helpful in identifying areas for improvement. We have conducted new experiments and analyses to address each of your concerns. We provide the details below, in the order of (L4 → L2 → L5 → L3 → L1):
>
> ***
> ## Evaluation on Open-Source Models
> > L4: The analysis could be broadened by including powerful open-source models to see if different failure patterns emerge.
>
> > **Action**: We will add the 400 newly generated MAS traces, LLM annotation, and the corresponding analysis to the final version of the paper. These traces will also be added to MAD on Hugging Face at the end of the rebuttal period (due to rebuttal policy).
>
> Inspired by your feedback, we have expanded our study to include two powerful open-source models. We generated and analyzed 400 new execution traces using `Qwen2.5-Coder-32B-Instruct` and `CodeLlama-7b-Instruct-hf` on the ChatDev and MetaGPT frameworks for the ProgramDev-v2 benchmark, and annotated with our LLM judge.
> The table below shows the number of occurrences for each failure mode across 100 traces for each configuration (for the 100 tasks in the ProgramDev-v2 benchmark).
>
> Table: Failure Mode Occurrences in 400 Traces with Open-Source Models
> | Failure Mode | Qwen ChatDev | Qwen MetaGPT | CodeLlama ChatDev | CodeLlama MetaGPT |
> |:-------------|-------------:|----------:|--------------:|-----------:|
> | 1.1          | 35           | 12        | 76            | 94         |
> | 1.2          | 4            | 1         | 45            | 12         |
> | 1.3          | 96          | 35        | 97            | 99         |
> | 1.4          | 1            | 0         | 46            | 23         |
> | 1.5          | 94          | 3         | 97            | 76         |
> | 2.1          | 2            | 0         | 50            | 9          |
> | 2.2          | 1            | 4         | 16            | 15         |
> | 2.3          | 9            | 0         | 76            | 57         |
> | 2.4          | 0            | 0         | 2             | 0          |
> | 2.5          | 2            | 12        | 42            | 40         |
> | 2.6          | 20           | 16        | 93            | 18         |
> | 3.1          | 1            | 47        | 25            | 26         |
> | 3.2          | 16           | 51        | 67            | 55         |
> | 3.3          | 12           | 32        | 69            | 56         |
>
> This new analysis reveals two key findings:
> 1. There is a significant performance difference between the two open-source models. `Qwen` is substantially more robust than `CodeLlama` on these tasks, exhibiting far fewer failures overall.
> 2. Both open-source models show a higher frequency of failures compared to the leading closed-source models analyzed in our paper (GPT-4o and Claude-3). This suggests a performance gap and highlights important areas for future improvement in open-source models for multi-agent tasks.
> Thank you for suggesting this important addition. By including open-sourced model traces with annotation, it also allows the community to explore model-level improvements to enable multi-agent systems.
>
> ***
>
> ## Inter-Annotator Agreement Clarification
> > L2: A concern was raised that the LLM annotator's agreement with human experts (κ=0.77) is not high enough leading to potential systemic bias in the final dataset.
>
> > **Action**: We will add clarification to the paper to better contextualize this agreement score.
>
> We agree that ensuring the reliability of the LLM annotator is crucial. While there is always room for improvement, we believe **our reported agreement score is strong** for this task for two key reasons.
>
> 1. Cohen’s Kappa is already a conservative metric that corrects for chance agreement.
> 2. Based on statistical literature [1], a Kappa score between between 0.4-0.75 considered “fair to good” and **above 0.75 is typically characterized as "excellent" agreement**. This is also pointed out by reviewers `npoU` and `G8Ce` that our score indicate high reliability and being recognized as a strength for rigorous annotation approach.
>
> [1] J. L. Fleiss, B. Levin, and M. C. Paik, *Statistical Methods for Rates and Proportions*, 3rd ed., Wiley, 2003.
>
> ***
>
> ## MAD Clarification
> > L5: the discrepancy in the number of released human-annotated traces, and the role of the 150 traces used for Grounded Theory.
>
> > **Action:** We will update the `MAD-human` dataset with the two missing traces after the rebuttal period and will clarify the numbers in the final paper.
>
> The 21 vs. 19 Released Traces: Thank you for bringing this to our attention. There should be 21 human-annotated traces in the MAD-human release. It is an oversight from our end. We have prepared the missing files and will add them to the public dataset on Hugging Face immediately following the rebuttal period to comply with conference policies.
>
> The 150 Traces: The 150 traces were used for our initial, exploratory analysis with Grounded Theory to *develop* the first version of the MAST taxonomy. These traces are part of the main MAD dataset, and not part of MAD-human. The MAD-human subset consists of independent annotations from three human experts *using MAST* gathered during our Inter-Annotator Agreement (IAA) studies to formally validate the taxonomy.
>
> ***
>
> ## Singular Root Cause
> > L3: while MAST provides an excellent classification of observable failures, the path to diagnosing a single root cause is a non-trivial challenge.
>
> > **Action:** We will clarify in the paper that MAST is designed to identify observable failures, which is the necessary first step before root cause analysis.
>
> We agree there is a crucial distinction in debugging complex systems: identifying an observable failure mode versus pinpointing its underlying root cause.
> MAST, by design, focuses on observable failure modes. By creating a systematic framework for observable failures, our work provides the necessary first step to guide the more complex diagnosis of the root cause.
>
> A useful analogy is a segmentation fault in traditional programming. The "`segfault`" is the observable failure, which is analogous to a failure mode in MAST like `1.1 Disobey Task Specification`. The underlying root cause, however, could be one of many implementation-specific issues, such as a null pointer dereference, a buffer overflow, or, in the case of MAS, a flawed prompt, a bug in an agent's role definition, or a model limitation, precisely as you outlined.
>
> Building a full-fledged agent debugger capable of automated root cause analysis is a new and important research goal that is beyond the scope of our current work, but is a very interesting future work and extension on MAST.
>
> ***
>
> ## The Evolving Nature of the MAST Taxonomy
> > L1: MAST, while thorough, may not be exhaustive given the rapid evolution of the MAS landscape.
>
> > **Action**: We have ongoing open-source effort to maintain MAST and will release additional open-source tools to support MAST’s evolution.
>
>
> We completely agree it is a challenge. Given rapid dynamic in agent, no single taxonomy can be perfectly exhaustive. Our goal with MAST was to provide the crucial **first** standardized and empirically grounded taxonomy for the community to build upon.
>
> We designed MAST to be both general and extensible, and we have multiple efforts underway to ensure it remains a living standard:
> 1. **Generalizability**. We validated the current taxonomy on two unseen frameworks, where our LLM-Judge achieved high inter-annotator agreement (Cohen's Kappa κ = 0.79) with human experts. This shows that MAST is already robust enough to capture failures in systems it was not trained on.
> 2. **Community Adoption and Extension**. We are encouraged that the community is already using and extending our work. For example, recent research from IBM on AssetOpsBench leverages MAST to categorize failures in their benchmark and then builds upon it by identifying new, domain-specific failure modes . This is exactly the kind of community-driven evolution we hoped to encourage.
> 3. **New Open-Source Tools**. To further support this, we are releasing an open-source Python library and an interactive dashboard. These tools will make it easier for other researchers to apply MAST to their own systems and propose extensions to the taxonomy.
>
> ***
>
> Thank you again for helping us improve the paper and hope our response can resolve your concerns! Please let us know if you have any further questions. We will be actively available for responses. If you feel your concerns are addressed, please consider re-evaluating our work. Looking forward to hearing from you :)
>
> Best,
>
> The Authors

---

> > ### Comment · Reviewer_yBzE · 2025-08-04
> > **Thanks for the rebuttal**
> >
> > The authors' rebuttal has addressed most of my concerns. I believe that this work opens an interesting direction towards evaluating and improving the performance of Multi-Agent LLM Systems. Therefore, I decide to increase my score. Good luck!

---

> > > ### Author Response · Authors · 2025-08-06
> > >
> > > Thank you so much for the score increase :) We are grateful for the recognition of the paper and the artifacts' usefulness to the community.

---

### Official Review · Reviewer_G8ce · 2025-07-02

**Rating:** 5
**Confidence:** 4

**Summary:**

The authors propose MAD, a public dataset of execution traces drawn from seven popular MAS frameworks, and derive MAST, a 14-mode taxonomy that groups failures into specification issues. This work also builds an LLM-as-a-Judge pipeline then scales these annotations with good reliability. MAST provides a systematic analysis on the capabilities of MAS frameworks, base models, and agentic tasks.

**Dataset Code Accessibility:**

Yes

**Dataset Code Comments:**

All the data and code for the key processes in this work is provided with documentation. The authors can further consider extracting the LLM-as-a-judge pipeline notebook into a program with good documentation.

**Ethical Considerations:**

No, there are no or only very minor ethics concerns

**Final Justification:**

The authors provide experiment results on my questions, which address most of my concerns. Considering the potential in using this benchmark as a guidance for future MAS design, I decide to increase my score for its benefits to the NeurIPS community.

**Limitations Weaknesses:**

1. This work considers more on the relationship between discovered failures and specific MAS frameworks. However, the failure distribution (Figure 4) may also differ from different benchmarks the agent executes. The benchmark difference (Table 1) may also result in different failure patterns, which should be further revealed.
2. The analysis on specific issues does not tie with the success of the task. Will these failure modes also be observed in successful patterns? In other words, will the success rate highly depend on avoiding these failure modes?
3. The authors should provide potential cost on using the LLM-as-a-judge pipelines to detect failure modes.

**Strengths Contributions:**

1. The paper delivers the first rigorous, empirical investigation of why large-language-model multi-agent systems (MAS) fail under specific tasks.
2. The LLM-as-a-judge-based evaluation help identify key issues in MAS traces with high reliability.
3. Analysis on MAD reveals high failure rates in current MAS frameworks and shows that prompt and topology tweaks can raise success by up to 15.6 %, highlighting system design—not just model choice—as a key lever
4. All artifacts—dataset, taxonomy, and annotator—are openly released, which is helpful to the community.

---

> ### Author Rebuttal · Authors · 2025-07-31
>
> Hi Reviewer G8ce,
>
> Thank you for your valuable time and thorough review. We appreciate you recognizing our work as the "**first rigorous, empirical investigation**" into why multi-agent systems fail, and for highlighting the reliability of our LLM Judge, the value of our open-source artifacts, and our analysis showing that system design is a key lever for improvement.
>
> Your questions and feedback help improve the depth of our analysis, and we are excited to share the results of our new analysis below, in the order of ( L2 → L1 → L3):
>
> ***
>
> ## Failures in Successful vs. Failed Tasks
> > L2: do failure modes also appear in successful tasks and how they relate to the final success rate ?
>
> > **Action**: we will include the added analysis below to the final paper
>
> Thank you for this interesting question. Inspired by your feedback, we conducted a new analysis on the ProgramDev benchmark to compare the failure profiles of successful versus failed execution traces on ChatDev and MetaGPT.
> The table below shows the frequency of each failure mode (normalized by the number of traces in each category (Success vs. Fail).
>
> Table: Frequency of Occurrence of Failure Modes in Successful/Failed Executions
> | Failure Modes         | 1.1   | 1.2  | 1.3   | 1.4  | 1.5  | 2.1  | 2.2  | 2.3  | 2.4  | 2.5  | 2.6  | 3.1  | 3.2  | 3.3  |
> |:----------------------|------:|-----:|------:|-----:|-----:|-----:|-----:|-----:|-----:|-----:|-----:|-----:|-----:|-----:|
> | ChatDev Successful    | 20.0% | 0.0% | 20.0% | 0.0% | 0.0% | 0.0% | 10.0%| 10.0%| 0.0% | 0.0% | 10.0%| 0.0% | 10.0%| 20.0%|
> | ChatDev Failed  | 25.0% | 0.0% | 20.0% | 5.0% |10.0% | 5.0% | 5.0% | 5.0% | 5.0% | 0.0% | 15.0%| 5.0% | 10.0%| 25.0%|
> | MetaGPT Successful    | 33.3% | 0.0% | 16.7% | 0.0% | 0.0% | 0.0% | 8.3% | 8.3% | 0.0% | 0.0% | 8.3% | 0.0% | 16.7%| 16.7%|
> | MetaGPT Failed  | 16.7% | 0.0% | 22.2% | 5.6% |11.1% | 5.6% | 5.6% | 5.6% | 5.6% | 0.0% |16.7% | 5.6% | 5.6% | 27.8%|
>
> This new analysis reveals three interesting findings:
> 1. **Successful runs are not failure-free**. Our results show that failures occur in both successful and failed traces, but failed traces have a higher overall frequency of failures. This confirms that a higher number of failures do signal a higher chance of final task failure.
> 2. **Some failures are more "fatal" than others**. The data shows a clear distinction between failure types. Certain failures, such as `1.5 Unaware of Termination Conditions` and `2.4 Information Withholding`, appear almost exclusively in failed runs, suggesting they are critical bugs that are highly likely to derail the task.
> 3. **Some failures are non-fatal**. In contrast, verification-related failures like `3.2 No or Incomplete Verification` and `3.3 Incorrect Verification` appear frequently even in successful runs. This suggests that while these systems can complete some tasks, their verification process still contains flaws. MAST identifies such systemic weaknesses, even when they do not cause an immediate task failure.
>
> ***
> ## Finer-Grain Failure Distribution:
> > L1: The failure distribution may also differ across different benchmarks, not just frameworks.
>
> > **Action**: We will add the benchmark specific distribution graph and analysis to the paper.
>
> This is an excellent point. The failure distribution is indeed dependent on the benchmark (i.e., the type of task) being evaluated, in addition to the framework and model. We will clarify this in the paper and add a new figure showing a breakdown by benchmark.
> In response to your feedback, we conducted a new analysis on data from our MAD dataset. We fixed the MAS framework (`AG2`) and the model (`GPT-4o`) while varying the benchmark. Due to rebuttal format limitations, we present the results in the table below. The failure rate is normalized by the number of traces.
>
> Table: Breakdown of FC Distribution on Benchmarks
> | MAS         | LLM        | Benchmark        |   FC1: Specification |   FC2: Inter-Agent Misalignment |   FC3: Verification Issues|
> |:-----------|:-----------|:-----------------|------:|------:|------:|
> | AG2        | GPT-4o     | GSM              |  0.53 |  0.33 |  0.37 |
> | AG2        | GPT-4o     | MMLU             |  1.06 |  1.01 |  0.6  |
> | AG2        | GPT-4o     | Olympiad         |  1.19 |  1.21 |  0.67 |
>
> We observed that the more challenging the benchmark is (e.g., Olympiad vs. GSM), the higher the failure rate. The distribution also changes significantly. While the failure profiles for MMLU and Olympiad are similar, the GSM benchmark results in a much lower number of `FC 2 - Inter-Agent Misalignment` and `FC 1 - Specification failures`.
>
> Beyond this paper, we are happy to share that we have an upcoming release of an open-source agent trace dashboard. This is a separate line of work that integrates MAST, and it will allow users to explore finer-grained failure distributions, including breakdowns by benchmark.
> ***
> ## LLM Judge Cost Breakdown
> > L3: report the judge cost in paper
>
> > **Action**: We have collected the cost breakdown and will do the final paper.
>
> Thanks for the suggestion. We have analyzed the API costs for our LLM-as-a-Judge pipeline across all traces in our study. The average cost across all MAS frameworks is **1.8 USD**, and the average cost per MAS highly depends on the length of the traces.
>
> The table below shows the breakdown:
>
> Table: Average Cost of LLM Annotator
> | MAS         | Average Cost Per Trace (USD) |
> |:--------------|-------------------:|
> | AppWorld      | 0.3740             |
> | HyperAgent    | 0.9695             |
> | AG2           | 1.1656             |
> | ChatDev       | 2.1272             |
> | MetaGPT       | 2.4455             |
> | MagenticOne   | 1.3056             |
> | OpenManus     | 4.1409             |
>
> ***
>
> **Thank you again for helping us improve the paper and hope our response can resolve your concerns! Please let us know if you have any further questions. We will be actively available for responses. If you feel your concerns are addressed, please consider reflecting this in their updated score. Looking forward to hearing from you. :)**
>
>
> Best,
>
> The Authors

---

> > ### Comment · Reviewer_G8ce · 2025-08-04
> >
> > I thank the authors for their response, which addresses most of my concerns. I encourage the authors to further develop this work on the following points:
> >
> > 1. How failure modes relate to success. From the new results in the rebuttal we can see that some failures are critical and some are not. May we get some insights on assigning different modes with different weights, or even providing a failure level (e.g., fatal, error, warning, etc., like a logging system)? I think the authors can provide some analysis on the importance of each failure type and elaborate on potential use.
> > 2. How each failure mode can be update-to-date with progressing MAS research. Your provided table shows that the failure rate of FC1-3 grows similarly with the difficulty, which is a nice proof of generalization. However, according to the feedback from other reviewers, the authors can consider providing more details on how the benchmark & dataset can adapt to other benchmarks and methods.
> > 3. Code with better documentation and examples. As I conveyed in the previous review, the authors should better finalize the code base, e.g., making the LLM-as-a-judge evaluation as a flexible module and documenting how to test traces generated with other benchmarks or methods.
> >
> > I welcome the authors to develop further projects according to this work, but I believe that these issues are highly related to this paper and should not be considered as part of future directions. Thus I encourage the authors to finalize this work according to these points and also other reviewers' comments.

---

> > ### Author Response · Authors · 2025-08-06
> >
> > Dear Reviewer G8ce,
> >
> > Thank you for your thoughtful and constructive follow-up. We are very happy to hear that our initial response addressed most of your concerns, and we appreciate the clear, actionable guidance on how to further strengthen our paper. We agree that the points you raised are highly related to this work, and we are committed to addressing them in the final version of the manuscript.
> > ***
> > ### 1. On the Relationship Between Failure Modes and Success
> >
> > Thank you for the suggestion to quantify failure criticality with weights or levels (e.g., fatal, error, warning). This touches on the important distinction between a diagnostic framework like MAST, which systematically identifies failure patterns, and a full-fledged debugger, which might assess the severity.
> >
> > We agree this is an important insight to include. In the final version of the paper, we will expand our discussion to explicitly introduce the concept of "critical" versus "non-critical" failures. We will use our new analysis—which shows that certain verification failures, for example, often appear even in successful tasks—as a concrete example to illustrate this point.
> >
> > Moreover, we do not plan to aggregate failure modes into a single composite score in this work, as introducing fixed weights would risk overgeneralization and misinterpretation. Because the "fatality" of a given failure mode is highly context-dependent (e.g. FM 3.2: No or Incomplete Verification can be fatal for a safety-critical application but non-fatal for a trivial code generation task), we believe the right approach is to provide transparent per-mode counts and let developers interpret severity in their own operational context. We will acknowledge this in the current paper and leave the design of a detailed scoring/weighting scheme for failure modes to future work, given the significant domain and application specific tradeoffs involved. Moreover, we want to emphasize (and will highlight this point further in the final version of the paper) that our primary contribution is providing the MAD dataset itself with over 1,000 traces and the standardized MAST taxonomy. We believe this foundational work is a crucial step that enables this emerging line of research into agent debuggers and observability. We also hope MAD and MAST will enable others to explore severity weighting and alternative scoring frameworks in a principled way.
> >
> > ***
> > ### 2. On Keeping the Taxonomy Up-to-Date and Adaptable
> >
> > This is a crucial point for ensuring the long-term value of our contribution. MAST is designed as a foundational, living taxonomy that can adapt as the MAS landscape evolves. We have taken a multi-pronged approach to ensure this:
> > - **Proven Generalizability**: We validated the current taxonomy on two unseen frameworks, where our LLM-Judge achieved high inter-annotator agreement (Cohen's Kappa κ = 0.79) with human experts. This demonstrates that MAST is already robust enough to capture failures in new systems.
> > - **Community Adoption**: We are encouraged that the research community and enterprise production testing are already using and extending our work. For example, recent work from IBM on `AssetOpsBench` leverages MAST to categorize failures in their benchmark and then builds upon it by identifying new, domain-specific failure modes. This is exactly the kind of community-driven evolution we hoped to inspire.
> >
> > ***
> > ### 3. On Code Documentation and Examples
> >
> > We appreciate your feedback here and are pleased to share that we have already taken steps to make our artifacts more flexible and user-friendly for the community:
> > - **Open-Source Tooling**: We have released an interactive dashboard to make it easier for other researchers to apply MAST to their own systems, validate our findings, and propose extensions to the taxonomy.
> > - We will release our annotation pipeline as a **Python package**, allowing users to analyze traces from any MAS framework with minimal configuration.
> > - The release will include best-practices documentation on how to structure MAS execution traces to maximize the utility of our analysis tools.
> >
> > ***
> >
> > **We believe these updates make clear our commitment to ensuring our contributions remain robust, extensible, and maximally useful to the MAS research community. Thank you again for your invaluable guidance in strengthening our paper. Please let us know if you have any further questions.**
> >
> > Thank you,
> >
> > Authors

---

> > > ### Comment · Reviewer_G8ce · 2025-08-08
> > >
> > > I thank the authors for providing the clarifications. I am generally satisfied with the rebuttal content and potential updates of this work. I will increase my rating.

---

### Official Review · Reviewer_npoU · 2025-07-03

**Rating:** 6
**Confidence:** 5

**Summary:**

The paper benchmarks and systematically explores failure modes of multi-agent systems. The paper introduces MAD — Multi-Agent Failure Detection dataset of more than 1000 annotated execution traces from 7 open source multi agent system frameworks. This dataset relies on Multi-Agent System Failure Taxonomy (MAST) built by analysis of 150 execution traces utilizing an elegant annotation pipeline combining human annotations with high inter-annotator agreement and an LLM as a judge developed to have high agreement with the human experts. The resulting MAST has 16 failure modes across three categories: specification issues, inter-agent misalignment, and task verification. The calibrated LLM judge is then used to annotate the full MAD dataset. The paper also provides practical recommendations for building multi-agent systems.

**Dataset Code Accessibility:**

Yes

**Dataset Code Comments:**

code provided

**Ethical Considerations:**

No, there are no or only very minor ethics concerns

**Final Justification:**

This paper is no doubt an impactful and valuable contribution to the community that is already being adopted in open source frameworks and other research works as the authors mentioned.

**Limitations Weaknesses:**

A few questions:
* Have you done any analysis of execution traces on the same tasks and beginning of conversation traces, but resampled conversation tree branches?
* You mention "We define failures as instances where the MAS does not achieve its intended task objectives.” How would MAST apply in more vague tasks where it is difficult to measure whether the objective has been achieved? E.g. creative tasks? How to use MAST to diagnose and improve agentic systems in that case?
* The paper mentioned an example for the practical value of MAST — is there a benefit to actually applying MAS vs just analyzing execution errors? Would the “CPO not waiting for the CEO approval” error be spotted by simply looking at the most common errors? Or would simply looking at execution errors not be informative enough?
* Is MAST able to guide testing of multi-agent systems? You mentioned that you do not claim that it covers every single failure mode, but would you be able to say that it covers the most common ones? That is, for a new multi-agent system, would coming up with test tasks to probe for error categories within MAST provide sufficient failure mode coverage? Or is it possible for a new multi-agent system to have a very pronounced and frequent failure mode not covered by MAST? The answer is likely yes for very different multi-agent systems, but what about multi-agent systems that are in scope of this study, like coding agents?
    * Can you recommend a way to automate creation of such a test task set based on MAST? Would your LLM as judge be able to engineer these test questions?
* Does the distribution of the failure modes depend on the multi-agent system design patterns/topology? Can you comment on the difference of the distribution of the failure modes across the 7 different MAS frameworks explored in your study?
* If MAST is used for root cause analysis of MAS failures, how high is the potential of conflating root causes leading to inaccurate diagnosis?

**Strengths Contributions:**

* Strengths:
    * The paper addresses a timely and critical problem of multi-agent system failure mode analysis. Production agentic systems often inevitably have to be multi-agent for efficient context window management and latency considerations and research with multi agent system design best practices is crucial.
    * Both the proposed taxonomy MAST along with the distribution analysis of the failure modes and the MAD dataset are comprehensive and valuable
    * Really good annotation approach — human experts have high agreement (κ = 0.88), the LLM-as-judge was developed to have high agreement with human experts on five MAS frameworks and then validated on two unseen ones achieving κ = 0.79
    * The findings diagnosing root causes of multi agent system failures across specification issues, inter-agent misalignment, and task verification provide both clear best practices, a debugging framework, and directions for further research
    * MAST provides a practical debugging and improvement tool — as shown in case studies, it can be used to pinpoint specific flaws and improve multi-agent systems

---

> ### Author Rebuttal · Authors · 2025-07-30
>
> Hi Reviewer npoU,
>
> Thank you for your valuable time, detailed review and strong score. We appreciate you recognizing that our work addresses a timely and critical problem with valuable contributions (MAST and MAD), a rigorous annotation methodology, and practical utility for guiding future research.
>
> We appreciate that you found no major weaknesses and instead raised 6 insightful questions. Your feedback has helped us improve the paper, and we provide our responses to each point below, in the order of (Q5, Q3, Q2, Q4, Q6&1).
> ***
> ## Fine-Grain Failure Mode Distribution
> > Q5: ​​are failure distributions affected by the multi-agent system's design and topology?
>
> > *TLDR*: Yes, failure distributions are dependent on both the system's architectural design and the underlying model. Our paper presents this in two ways, and in response to your question, we conducted an additional analysis to explore this relationship further.
>
> > **Action**: We will add the analysis and the new breakdown to the final version of our paper.
>
> **Observation**: As shown in Figure 4 of the paper, the overall failure distributions vary across frameworks. For example, both AppWorld and MagenticOne show a high rate of `FM 3.1 - premature termination` failures. A common architectural pattern in both is a dynamic orchestrator-worker topology, where the orchestrator agent picks the worker agent at runtime based on the task. We hypothesize that this design, while giving the orchestrator agent more agency, has a higher risk of losing track of all the partial tasks compared to MAS with a more rigid, predefined workflow.
>
> **Controlled Topology Comparison**: To provide a more controlled comparison, we analyzed two systems (MetaGPT and ChatDev) on the same benchmark (ProgramDev-v2) and model (GPT-4o). As detailed in Appendix E Figure 9, this comparison points out the importance of design differences in MAS topologies and architecture on failure mode distributions. In particular, MetaGPT has a more rigid and comprehensive system design where they provide SOPs to different agents, which is reflected by the very small number of `Failure Category 1- specification issues` and `Failure Category 2 - inter-agent misalignment` occurrences compared with ChatDev. On the other hand, ChatDev has a stronger emphasis on verification: there are multiple stages of testing, such as a code reviewer agent and a system testing agent, and the review process across different hierarchies of agents (the CEO Agent, the CTO Agent, the software developer agent) seem to help with the `Failure Category 3 - verification` related failure modes a lot, when compared to MetaGPT, that has a weaker verification procedure where the unit tests seem to be superficial and missing edge cases most of the time.
>
> To further investigate your question, we conducted a **new analysis** on our MAD dataset to isolate the effects of the number of agents, the base model, and the benchmark. The table below presents a summary of the analysis. The results suggest that the number of agents *does not have a direct correlation* with failure rates. However, the choice of the base LLM (e.g., GPT-4o vs. Claude for AG2) and the benchmark can change failure rates by more than 2x, showing they are often more critical factors.
>
> Table: Average Failure Count per Trace by Category
> | MAS        | Max Num Agents | LLM        | Benchmark        |   FC1: Specification |   FC2: Inter-Agent Misalignment |   FC3: Verification |
> |:-----------|-----------:|:-----------|:-----------------|------:|------:|------:|
> | AG2        | 2 | Claude | GSM |  1.11 |  1.12 |  0.65 |
> | AG2        | 2 | GPT-4o | GSM  |  0.53 |  0.33 |  0.37 |
> | AG2        | 2 | GPT-4o | MMLU |  1.06 |  1.01 |  0.6  |
> | AG2        | 2 | GPT-4o | Olympiad |  1.19 |  1.21 |  0.67 |
> | OpenManus  | 5 | GPT-4o | ProgramDev |  0.53 |  0.33 |  0.37 |
> | MagenticOne| 5 | GPT-4o | GAIA |  0.98 |  0.89 |  0.55 |
> | ChatDev    | 5 | GPT-4o | ProgramDev |  0.94 |  0.7  |  0.52 |
> | MetaGPT    | 5 | Claude | ProgramDev |  1.06 |  0.81 |  0.57 |
> | MetaGPT    | 5 | GPT-4o | ProgramDev |  0.94 |  0.7  |  0.52 |
> | AppWorld   | 11 | GPT-4o | Test-C |  0.53 |  0.33 |  0.37 |
>
> ***
> ## Abstraction
> > Q3: the practical benefit of applying MAST in debugging as compared to analyzing raw execution errors.
>
> > TLDR: MAST provides a necessary level of *abstraction*. This abstraction creates a shared terminology that enables scalable detection and comparison analysis. MAST is now used in a few production agent development pipelines.
>
> Analyzing MAS traces presents a trade-off. Manually inspecting raw execution traces to look out for every instance of error is not scalable and challenging as traces are long and highly diverse. On the other end, a pass/fail metric is overly simple for development. MAST provides an intermediate level of abstraction that offers three key benefits:
> 1. **A shared terminology for analysis**. MAST’s list of 14 common failure modes enables scalable, automated analysis with the LLM-Judge pipeline. Not only for the LLMs, MAST also improves communication for human developers. This is a feedback that we got from MAST’s industry user building production agent, that the engineering teams uses MAST to detect and communicate failure more efficiently, which boosted their productivity.
> 2. **Enable comparison across frameworks**. MAST abstracts away framework-specific errors like  "CPO not waiting for CEO approval", yet captures the generalizable pattern behind it (e.g., FM 2.1: Disobey Role Specification). This enables comparisons between different system architectures.
> 3. **Guidance for future research**. By identifying and categorizing prevalent errors, MAST offers clear insights into the common challenges in MAS development today, guiding future research for ourselves and the community.
> ***
> ## Creative Tasks
> > Q2: how MAST could be applied to more subjective or creative tasks where success is hard to measure?
>
> This is a great question that touches on an important area for future research. For this foundational study, we deliberately focused on tasks with objective success criteria to establish a clear, empirical baseline for failure analysis. Enabling agents on unverifiable or hard-to-verify tasks is an emerging research direction. We agree that extending this analysis to creative domains is a valuable next step.
> ***
> ## MAST for Guiding and Automating Testing
> > Q4: whether MAST can guide automated testing and if it sufficiently covers the most common failures?
>
> > TLDR: Yes, MAST is able to guide testing. Its utility on new systems was validated in our paper. Recent research works are using MAST. And we are releasing new open-source tools to make automated failure analysis easier for the entire community.
>
> > **Action**: We will release the dashboard and link to the paper.
>
> **Effectiveness on new systems**. To ensure MAST was not specific to the frameworks used for its creation, we validated it on two unseen systems and unseen benchmarks - MagenticOne and OpenManus with GAIA and MMLU. Our LLM-Judge, achieved a high inter-annotator agreement (`Cohen's Kappa = 0.79`) when compared with human experts annotation on these new frameworks.
>
> **MAST is the first yet living taxonomy**. You are correct that a completely new system paradigm might introduce failure modes not yet captured in MAST. We do not claim our current 14 modes are exhaustive. Instead, we view MAST as the crucial first step toward a standardized understanding of agent failures. We are committed to an open-source effort where the community can contribute to and expand the taxonomy over time.
>
> For example, recent work from IBM on *AssetOpsBench* leverages MAST to categorize failures in their benchmark. They then build upon our work by identifying new, domain-specific failure patterns, demonstrating exactly the kind of community extension we envision.
>
> **We are releasing new tools to support automated analysis**. To support the use case you described, we are releasing an open-source Python library (with a CLI/API) and an interactive failure analysis dashboard website. These tools will allow researchers to input their own execution traces and automatically receive a full failure analysis based on the MAST categories. While our current tools focus on analyzing existing traces, we agree that using the LLM-Judge to automatically generate new test cases is a fascinating direction for future work.
>
> ***
> ## MAS Debugger
> > Q6 & Q1: MAST's role in root cause analysis and analyzing modified or resampled execution traces.
>
> These questions touch on the critical and broader topic of debugging multi-agent systems.
>
> You raise an important distinction regarding root cause analysis. MAST is designed as a diagnostic framework to systematically identify and categorize failure patterns. While this is a crucial input for any debugging process, we agree that a full-fledged diagnostic tool, i.e. a debugger, capable of identifying root cause (ie, how a failure occur, localization, etc..) analysis is a distinct and important research goal that is beyond the scope of our current work. Our contribution is to provide a foundational component that enables this emerging line of research.
>
> Your question about resampling traces highlights one of the exciting capabilities such a future MAS debugger / observability tool could have. The ability to intervene in an execution trace, analyze counterfactuals, and understand system behavior is a key problem in agent observability. MAST provides a standardized language to describe the outcomes of such experiments.
> This new line of research direction is something that we are actively exploring, and we appreciate you highlighting its importance.
> ***
> Thank you again for your valuable time and insightful feedback. We are happy to address any additional questions, and we appreciate your strong support and positive recognition of our paper.
>
> Best,
>
> The Authors

---

### Comment · Area_Chair_K9Qe · 2025-08-08

Dear Authors and Reviewers,

The deadline for author-reviewer discussions is approaching. If you haven’t done so already, please ensure there are sufficient discussions for both the submission and the rebuttal.

Reviewers, please make sure you complete the mandatory acknowledgment AND respond to the authors’ rebuttal, as requested in the email from the program chairs.

Authors, if you feel that any results need to be discussed and clarified, please notify the reviewer. Be concise about the issue you want to discuss.

Your AC

---

### Note · Authors · 2025-08-16

**We would like to sincerely thank the reviewers, ACs, and SACs** for their careful and constructive evaluation of our work. We are grateful that the reviewers recognized multiple strengths of our work, which we summarize and categorize here into detailed contributions and broader research impact.

## Detailed contributions:

- Both the proposed taxonomy MAST and the MAD dataset were acknowledged as **valuable**, **comprehensive**, and **invaluable resources** crucial for the multi-agent system (MAS) research community.
- Our annotation approach was praised for its **rigor** and **reliability**.
- The LLM-as-a-judge evaluation framework effectively identifies key issues within MAS traces with **high reliability**, enabling systematic diagnosis of failures.
- The use of Grounded Theory combined with iterative Inter-Annotator Agreement studies ensured that the MAST taxonomy is **empirically grounded** and **robustly defined**.

## Broader research perspective:

- **The paper addresses a timely and critical problem** in the MAS field: a systematic and principled failure mode analysis that offers best practices for MAS designers and focused directions for future investigation.
- MAD and MAST serve **not only as an analytical taxonomy** but, together with the LLM annotator, also as a **practical improvement tool**. As demonstrated by our case studies, it can pinpoint flaws in MAS designs and workflows, enabling targeted mitigation strategies.
- We are releasing an **open-source Python library** and an **interactive dashboard**. These tools will make it easier for other researchers to apply MAST to their own systems and propose extensions to the taxonomy.

## Refinements thanks to reviewer discussion:

The concerns and suggestions raised by the reviewers during the discussion phase enabled us to enhance the presentation and potential impact of this work. For example, we studied how specific failure modes affect the task success rates, and broadened our evaluation by including several other open-source models. **We are pleased to have thoroughly addressed all the points raised by the reviewers**, and we are especially encouraged that at least **two reviewers chose to increase their scores** following the rebuttal phase. Moreover, **the other two reviewers acknowledged our clarifications without requesting further explanations**, from which we deduce that all points have been satisfactorily addressed.

Thank you again for your thoughtful consideration.

---

### Decision · Program_Chairs · 2025-09-18

**Decision:**

Accept (spotlight)

**Comment:**

All reviewers have provided positive scores for this submission, highlighting its strengths and key contributions. Given the unanimous positive feedback and the recognition of its contribution to the area, the AC carefully reviewed the paper and concurred with the reviewers' assessments, therefore supporting the decision to accept this submission.